# Fine-Tuning Arabic Large Language Models for improved multi-turn dialogue: A blueprint for synthetic data generation and benchmarking

Ahmed Mahmoud Misbah [1]*, Mohamed Farouk[2], Mustafa AbdulAzim[1]

1 College of Computing and Information Technology, Arab Academy for Science, Technology and Maritime Transport, Cairo, Egypt, 2 College of Computing and Information Technology, Arab Academy for Science, Technology and Maritime Transport, Alexandria, Egypt

* a.misbah5156@student.aast.edu

## Abstract

The rapid evolution of Large Language Models (LLMs) has fueled increasing interest in developing Arabic conversational systems capable of sustaining coherent multi-turn dialogues. However, progress remains constrained by the scarcity of large-scale, diverse, and high-quality datasets specifically designed for Arabic multi-turn interaction. This study presents a reproducible methodology for constructing such a dataset through structured prompting of an instruction-tuned Arabic LLM (Jais-13b-chat), yielding 43,316 multi-turn conversations across 93 topics and 151 countries. Two pre-trained Arabic language models (ArabianGPT-08B-V2 and AraGPT2-mega) were fine-tuned on this synthetic data and benchmarked against multilingual instruction-tuned baselines using a comprehensive evaluation framework combining automatic metrics (Perplexity and RAVEN) with structured human evaluation. Fine-tuned ArabianGPT-08B-V2 achieved the highest RAVEN score (0.823) for cross-model comparison, outperforming both fine-tuned AraGPT2-mega and instruction-tuned baselines while maintaining strong within-model perplexity (9.4). Human evaluation by two independent raters demonstrated acceptable inter-rater reliability (Cohen's κ = 0.229–0.739) with positive rank correlations (Spearman ρ = 0.424–0.759), yielding overall quality scores of 4.04–4.34 on a five-point scale. These findings demonstrate that high-quality, LLM-generated synthetic data effectively improves Arabic conversational models, providing a scalable, resource-efficient blueprint for dialogue systems in low-resource and culturally specific settings.

## Introduction

The emergence of ChatGPT in late 2022 marked a significant advancement in conversational artificial intelligence, demonstrating unprecedented capabilities in natural language understanding and generation [1]. This development catalyzed widespread adoption of transformer-based large language models (LLMs) for dialogue systems,

**Data availability statement:** "All relevant data are within the paper and its Supporting information files. Dataset has been provided at this link: https://doi.org/10.5281/zenodo.17855012".

**Funding:** The author(s) received no specific funding for this work.

**Competing interests:** The authors have declared that no competing interests exist.

including Google's Gemini, Meta's LLaMA, Mistral, and Falcon. These models represent a fundamental departure from earlier rule-based and retrieval-based chatbot architectures in their ability to leverage massive pre-training corpora and generalize across diverse natural language processing tasks, including machine translation, text summarization, and sentiment analysis. Most notably, modern LLMs maintain contextual coherence across extended multi-turn conversations, a capability that distinguishes them from previous generations of conversational systems.

Despite these advances, initial deployments of mainstream LLMs exhibited limited multilingual support, particularly for morphologically rich languages such as Arabic. This limitation has motivated the development of dedicated Arabic language models. While several Arabic pre-trained models exist, including AraBERT [2], AraGPT2 [3], and AraT5 [4], most were designed for general language understanding tasks rather than conversational interaction. Consequently, relatively few models have been specifically adapted for Arabic dialogue systems [5–7], and the development of robust multi-turn conversational models remains constrained by a critical resource bottleneck: the scarcity of large-scale, high-quality Arabic dialogue datasets.

More recently, instruction-tuned Arabic language models such as Jais-13b-chat [8] have emerged as promising foundations for dialogue systems. These models are trained on substantial multilingual corpora containing billions of Arabic tokens and subsequently fine-tuned using instruction-following datasets to improve alignment with human communicative intent. While they demonstrate competence in single-turn interactions and basic multi-turn exchanges, they are not explicitly optimized for maintaining coherence across extended open-domain conversations. This limitation becomes particularly apparent in dialogues requiring sustained context tracking and topical continuity, highlighting the need for task-specific adaptation to enhance their multi-turn conversational capabilities.

The remainder of the manuscript is organized as follows. The specific challenges posed by existing Arabic dialogue datasets and the research gap they create are formalized in the "Problem Statement" subsection, followed by an overview of the principal contributions of this study. Section 2 reviews related work on Arabic conversational AI and dialogue dataset construction. Section 3 details the methodology, including synthetic data generation procedures, data quality assurance measures, and model fine-tuning protocols. Section 4 presents comprehensive benchmarking results using both automatic metrics and human evaluation. Section 5 provides an in-depth analysis of model performance. Finally, Section 6 synthesizes the key findings, discusses implications for Arabic NLP research, and identifies directions for future work.

## Problem statement

The existing landscape of Arabic dialogue datasets presents significant challenges for model development. The JANA dataset [9] represents the only publicly available resource specifically designed for Arabic multi-turn conversation, yet its utility is constrained by both scale (3,000 conversations) and domain specificity (call center interactions). Other prominent Arabic NLP datasets, including ArabicaQA [10],

InstAr-500k [11], and Arabic-SQuAD [12], while larger in scale, suffer from systematic limitations that compromise their effectiveness for dialogue modeling. First, many are derived from English sources through machine translation, inherently failing to capture the linguistic subtleties and cultural contexts intrinsic to authentic Arabic discourse. Second, they are predominantly structured for extractive single-turn tasks such as question answering, rather than generative multi-turn dialogue. Third, datasets sourced from social media platforms often contain heterogeneous dialectal variations, informal register, and inconsistent content quality. Finally, manual annotation by human experts, while yielding high-quality data, remains prohibitively resource-intensive and limits achievable dataset scale.

These constraints collectively prevent the development of Arabic language models capable of maintaining contextual coherence over extended interactions in open-domain conversational scenarios. To address this gap, this study proposes a methodology for constructing a large-scale synthetic Arabic dialogue dataset through structured prompting of instruction-tuned language models. The utility of this synthetic data is then demonstrated through fine-tuning of pre-trained Arabic language models and evaluation of their performance on multi-turn conversational tasks. This approach builds on successful precedents in English NLP, including InstructWild [13] and Self-Instruct [14], which have demonstrated that LLM-generated synthetic data can effectively approximate human-generated content across various natural language tasks. Beyond replicating surface-level linguistic patterns, synthetic data generation offers several methodological advantages: it is substantially more cost-effective than human annotation, enables rapid iteration and dataset expansion, and allows systematic control over domain coverage and stylistic variation. These characteristics make synthetic data generation particularly valuable for under-resourced languages where large-scale human annotation is impractical.

**Novelty of work**

This study addresses the data scarcity challenge for multi-turn Arabic dialogue through four principal contributions:

1. **Large-Scale Synthetic Arabic Multi-Turn Dataset:** A unique dataset comprising 43,316 multi-turn Arabic conversations was generated through structured prompting of an instruction-tuned Arabic language model (Jais-13b-chat). Unlike existing Arabic dialogue datasets that are constrained by small scale (e.g., JANA's 3,000 conversations) or narrow domain coverage (e.g., call center interactions), the synthetic dataset spans 93 diverse topics across 151 countries, providing broad coverage of open-domain conversational contexts with extended dialogue length (mean: 14 turns, range: 5–111 turns).

2. **Reproducible Methodology for Controlled Synthetic Data Generation:** A comprehensive technical framework for generating high-quality synthetic multi-turn dialogue data is presented, detailing the systematic application of prompt engineering and hyperparameter optimization to control conversational diversity, contextual depth, and linguistic quality. The methodology addresses specific challenges in Arabic dialogue generation, including cultural appropriateness and maintenance of multi-turn coherence. Complete documentation of the generation pipeline enables replication and adaptation to other languages and domains.

3. **Empirical Validation through Fine-Tuning:** The practical utility of synthetic training data is demonstrated through fine-tuning of two pre-trained Arabic language models. Evaluation results show that models trained on synthetic data achieve performance comparable to or exceeding larger multilingual instruction-tuned baselines on multi-turn conversational tasks, establishing a resource-efficient pathway for developing specialized Arabic dialogue systems.

4. **Comprehensive Evaluation Framework for Arabic Multi-Turn Dialogue:** A novel benchmark specifically designed for assessing Arabic conversational systems in multi-turn settings is introduced. The framework integrates automatic metrics with structured human evaluation across multiple quality dimensions (fluency, relevance, diversity). This evaluation protocol addresses the gap in rigorous assessment tools for Arabic dialogue systems and provides a foundation for future comparative studies.

## Literature review

This section situates the present study within the broader landscape of Arabic conversational AI research, examining the evolution of chatbots, the current state of Arabic chatbot development, and the methodological approaches employed in constructing training datasets. Particular attention is given to the persistent resource constraints that have impeded progress in Arabic multi-turn dialogue modeling and the emerging role of synthetic data generation as a viable solution to these challenges.

### Chatbots

The early 2000s witnessed a fundamental transformation in chatbot architectures, progressing from rudimentary pattern-matching systems to sophisticated neural dialogue models. Early chatbots relied on hand-crafted rules and template-based response generation, but the integration of Machine Learning (ML) and Natural Language Processing (NLP) techniques enabled more adaptive systems capable of learning from interaction data. Commercial deployments such as IBM Watson, Apple's Siri, and Amazon's Alexa exemplified this transition, implementing hybrid architectures that combined retrieval-based mechanisms with increasingly capable generative components. Within this paradigm, two broad categories of chatbots emerged: open-domain chatbots designed for unrestricted, general-purpose interaction, and closed-domain chatbots optimized for specific applications such as customer support or technical assistance.

The advent of transformer-based Large Language Models (LLMs) represented a qualitative advancement in chatbot capabilities. Unlike their predecessors, transformer architectures effectively model long-range dependencies in sequential data, enabling the generation of fluent, contextually coherent text across extended conversational exchanges. The Generative Pre-trained Transformer (GPT) family of models demonstrated that pre-training on massive text corpora yields representations sufficiently rich to support human-like text generation across diverse domains. This architectural foundation enabled the development of multi-turn chatbots capable of maintaining topical coherence and contextual awareness across extended dialogue turns, a capability that had previously eluded rule-based and retrieval-based approaches.

The success of GPT-based chatbots catalyzed widespread development of transformer-based dialogue systems. Microsoft's DialoGPT [15], fine-tuned on Reddit conversational data, demonstrated that GPT-2 could be effectively adapted for open-domain dialogue generation. OpenAI's ChatGPT, initially based on GPT-3.5 [16] and subsequently evolved to GPT-4o [17], established new benchmarks for conversational fluency and instruction-following capability. Concurrently, alternative transformer architectures emerged with comparable dialogue capabilities: Google's Gemini [18], Meta's LLaMA [19], Mistral's Mixtral [20], and others [21] each demonstrated that instruction-tuning on diverse conversational corpora yields models capable of sophisticated multi-turn interaction. Collectively, these developments have established transformer-based LLMs as the dominant paradigm for contemporary chatbot development.

### Arabic Chatbots

Despite the rapid advancement of chatbots for English and other high-resource languages, Arabic chatbot development has proceeded at a markedly slower pace. This disparity stems from two interrelated factors: the scarcity of large-scale Arabic training data and the inherent linguistic complexity of Arabic, which encompasses substantial dialectal variation alongside Modern Standard Arabic (MSA). Consequently, Arabic chatbots remain fewer in number and generally inferior in performance relative to their English counterparts.

Early Arabic chatbots relied predominantly on rule-based architectures with limited generalization capacity. ArabChat [22] employed pattern-matching techniques to generate responses to user queries, while BOTTA [23] implemented retrieval-based response selection specifically for Egyptian Arabic. Domain-specific applications such as Ollobot [24] demonstrated the feasibility of rule-based Arabic dialogue for constrained tasks like health tracking. However, a comprehensive survey by AlHumoud et al. [25] concluded that Arabic chatbot development remained nascent, fundamentally

constrained by insufficient training resources. Some researchers attempted to circumvent data limitations through translation-based approaches: Mozannar et al. [12] developed SOQAL, a question-answering system leveraging translated resources, while Hajj et al. [26] explored sequence-to-sequence architectures for Arabic response generation.

The introduction of transformer-based Arabic language models substantially advanced the field. Antoun et al. [27] adapted the AraBERT architecture for conversational applications, while Al-Yahya et al. [6] developed AraConv using the multilingual mT5 model. Meshrif's ArRASA system [7] incorporated the DIET (Dual Intent and Entity Transformer) architecture to enhance intent recognition and entity extraction in Arabic dialogue. These models demonstrated marked improvements over rule-based predecessors in both response quality and domain coverage.

More recently, the emergence of multilingual instruction-tuned models has further transformed Arabic conversational AI. Jais-Chat [8], trained on extensive multilingual corpora containing substantial Arabic content, represents a new generation of models that address the English-centric limitations of earlier architectures. These systems exhibit enhanced linguistic coverage and instruction-following capabilities, enabling more sophisticated dialogue management and broader query handling. Nevertheless, while multilingual models provide improved baseline performance for Arabic, they are not explicitly optimized for extended multi-turn coherence, suggesting continued need for task-specific adaptation using dedicated Arabic dialogue resources.

## Arabic datasets for Chatbots

The rise of transformer-based pre-trained large language models (LLMs) has driven substantial progress in Natural Language Processing (NLP), particularly in conversational AI and chatbot development. These models are typically fine-tuned on specialized datasets tailored to specific downstream tasks. Early fine-tuning efforts primarily relied on monolingual, task-oriented datasets, such as English question-answering corpora [28]. Over time, however, the field has evolved toward more diverse and complex datasets, including multilingual corpora, instruction-tuning datasets, open-domain conversational data, and multi-turn dialogue datasets [29].

Non-task-oriented datasets are particularly important for training models to engage in multi-turn conversations, where interaction is not bound to a specific goal but aims to emulate natural, open-ended dialogue [30,31]. These datasets enable models to maintain context across multiple conversational turns, making them essential for building robust chatbots.

While significant advancements have been made in English and other high-resource languages, the development of Arabic-specific datasets remains limited. Existing Arabic resources include datasets such as JANA [9], Arabic-SQuAD [12], ArabicaQA [10], and InstAr-500k [11]. However, these datasets suffer from several limitations that hinder their effectiveness for training Arabic multi-turn conversational chatbots. For instance, JANA is restricted to a narrow domain (call centers) and contains only 3,000 dialogues. Others, like Arabic-SQuAD and ArabicaQA, are primarily designed for single-turn tasks or are based on translated content, which introduces linguistic and cultural noise.

This underdevelopment of Arabic datasets presents a set of specific challenges that must be addressed to advance Arabic conversational AI:

1. **Scarcity of High-Quality Multi-Turn Datasets:** Unlike English, where large-scale datasets for multi-turn conversational AI are abundant, Arabic datasets are limited in both quantity and scope. The JANA dataset [9], one of the few designed for Arabic multi-turn chatbots, is small, containing only 3,000 call center dialogues. Its limited size and narrow focus restrict its applicability to broader conversational contexts.

2. **Reliance on Translations:** Many Arabic datasets are translated versions of English datasets, such as Arabic-SQuAD [12]. While translations offer a starting point, they fail to capture the cultural and linguistic nuances of Arabic, including dialectal variations. Additionally, machine translation often introduces inaccuracies, further diminishing the quality of these datasets.

3. **Single-Turn Focus:** Most available Arabic datasets, such as ArabicaQA [10] and InstAr-500k [11], are designed for single-turn tasks like question answering. These datasets do not adequately support training models for open-ended, multi-turn conversational interactions.

4. **Issues with Data Sources:** Some Arabic datasets are sourced from informal platforms like social media and forums. While these sources offer diverse conversational data, they often include problematic elements such as profanity, biased content, and irrelevant or out-of-context responses, which can adversely affect the performance of chatbots.

5. **Dialectal Complexity:** Arabic is characterized by a rich variety of dialects that differ significantly from Modern Standard Arabic (MSA). Existing datasets often fail to represent these dialects comprehensively, limiting the ability of chatbots to understand and respond to users in their preferred form of Arabic.

6. **Manual Annotation Limitations:** Many Arabic datasets are created through human annotation, which is a time-consuming process that often results in small datasets. This restricts their usefulness for training large-scale conversational models.

These limitations underscore the urgent need for scalable, culturally grounded, and linguistically diverse Arabic datasets, particularly those tailored for multi-turn dialogue, in order to unlock the full potential of Arabic conversational AI.

## Dataset construction methods

To overcome the limitations outlined in the previous sub-section, researchers have explored various approaches for constructing datasets suitable for training multi-turn conversational models:

1. **Human-Crafted Datasets:** These are manually curated by annotators following specific guidelines to ensure high quality and contextual relevance. However, the manual nature of this process limits scalability and results in small dataset sizes [32–34].

2. **Mining Publicly Available Conversations:** Conversations from movies [34], TV programs, instant messaging [9], online forums, and social media platforms can provide a rich source of natural dialogue. While this approach offers diverse data, it introduces challenges such as copyright issues, cleaning profanity, handling dialectal diversity, and managing inaccuracies from speech-to-text transcription when processing audio data.

3. **Synthetic Datasets:** Synthetic datasets are generated by algorithms rather than being manually curated, providing an efficient solution to address data scarcity. These datasets can be produced using large language models (LLMs) [13,14]. LLMs such as GPT-4o [17], LLaMA [19], Gemini [18], Mixtral [20], Falcon [35], and Jais [8] have showcased their ability to create realistic and diverse conversational patterns by generating synthetic text.

Empirical validation of synthetic data for English multi-turn dialogue has yielded encouraging results. The PLACES framework [36] employs structured prompting techniques to synthesize social conversations, generating multi-turn dialogues that closely approximate human interaction patterns; models fine-tuned on PLACES data perform comparably to those trained on human-collected corpora. The Ultrachat dataset [37] similarly leverages LLM generation to produce diverse multi-turn conversations spanning varied topics and named entities, facilitating the development of models with enhanced coherence and contextual relevance. These studies collectively establish that LLM-generated synthetic data can effectively enhance multi-turn chatbot capabilities in English.

The application of synthetic data generation to Arabic dialogue remains nascent. ALMutairi et al. [38] demonstrated the feasibility of generating synthetic Arabic medical dialogues from patient notes, addressing resource scarcity in a specialized domain. The Arabic Stable LM 1.6B model [39] incorporated synthetic instruction-tuning data to improve Arabic benchmark performance, though this data was designed for single-turn instruction-following rather than multi-turn

conversation. While these studies affirm the value of synthetic data for Arabic language model adaptation, neither addresses the specific challenge of generating or fine-tuning models for open-domain multi-turn dialogue, the precise gap that the present study seeks to address.

## Methods

To investigate the effectiveness of LLM-generated synthetic data in enhancing Arabic conversational models, this study adopts a practical methodology centered on fine-tuning (illustrated in Fig 1). The proposed approach begins by generating a synthetic dataset composed of multi-turn Arabic dialogues using a capable instruction-tuned LLM. This dataset serves as the foundation for fine-tuning two pre-trained Arabic language models, aiming to improve their ability to engage in natural, dialogue-based interaction. The fine-tuned models are evaluated using a novel benchmark proposed in this study for assessing Arabic multi-turn chatbots, incorporating both quantitative and qualitative evaluation metrics to measure key aspects of conversational quality.

### Instruction-tuned LLM selection

A critical first step in the synthetic dataset generation pipeline is the selection of a suitable instruction-tuned LLM. Since the quality, coherence, and relevance of the generated dialogues are directly influenced by the capabilities of the model used for generation, the proposed approach follows a set of well-defined criteria for evaluating candidate models.

The first and most essential criterion is language support. The selected LLM must exhibit strong Arabic language capabilities, evidenced by training on a substantial corpus of Arabic text. Ideally, Arabic should constitute a significant portion of the model's training data to ensure proficiency in generating contextually appropriate and linguistically accurate Arabic. Models with robust multilingual capabilities that include significant Arabic training are also considered.

The second criterion pertains to model architecture. The selected LLM's architecture should represent the current state-of-the-art in LLM design. This includes, but is not limited to, transformer-based architectures with a large number of parameters (e.g., models with a scale comparable to or exceeding GPT-3), and advanced architectural innovations such

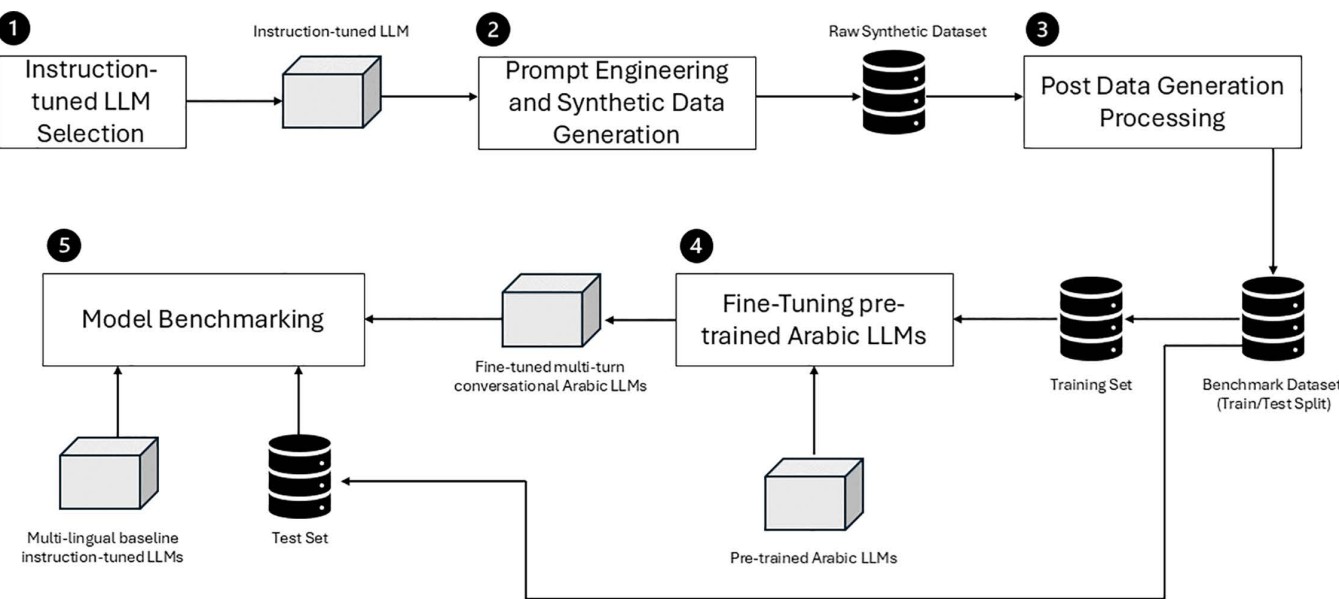

**Fig 1. Overview of the methodology for generating the synthetic Arabic dataset and fine-tuning pre-trained Arabic language models to multi-turn conversational Arabic LLMs.**

as Mixture-of-Experts (MoE) layers. These architectures have demonstrated superior capacity for language modeling and complex task execution.

Third, the size of the model's training dataset is considered. The training dataset should be of substantial size, containing at least 1 billion tokens. A large training corpus is essential for capturing the statistical regularities of language and enabling effective generalization. However, dataset quality and diversity are considered equally important alongside size.

Fourth, licensing and accessibility are key considerations. The model should be available under a license that permits its use for research, dataset generation, and the use of its generated output to train other models. Open-access models or those with permissive licenses are preferred to facilitate reproducibility, and legal use in downstream tasks. Availability on platforms like Hugging Face is a significant advantage, as it simplifies model access and deployment.

Finally, computational resource requirements are considered. Computational resource requirements must be feasible for execution within the constraints of readily available cloud computing platforms. Specifically, the model should be runnable Google Colab Pro+ instance equipped with an NVIDIA A100 GPU, 40 GB of GPU RAM, and 84 GB of System RAM. Models exceeding these resource limits are excluded from consideration to ensure accessibility and cost-effectiveness of the dataset generation process.

Based on these criteria, several instruction-tuned LLMs available as of mid-2024 were compared, as shown in Table 1.

After evaluating the candidates, Jais-13b-chat was selected as the most suitable model for this research. Its specialization in Arabic and English, open-source license, and strong multilingual performance made it a compelling choice. Its demonstrated fluency in Arabic generation, along with efficient inference requirements, enabled us to generate high-quality dialogue on accessible hardware without compromising linguistic richness or contextual coherence.

## Prompt engineering and synthetic data generation

The primary goal of this stage was to generate a high-quality Arabic dataset suitable for fine-tuning LLMs on multi-turn dialogue tasks. This dataset was envisioned to comprise fluent, coherent, and culturally relevant conversations while maintaining topical relevance within diverse scenarios across various countries. To ensure both cultural and topical diversity, this study's objective was to generate dialogues spanning 93 distinct topics (e.g., transportation, education, health, etc.) across 151 countries, with each conversation reflecting local contexts. To achieve this, two key strategies were employed: meticulous prompt engineering to guide the language model's output and careful tuning of the LLM's generation hyperparameters to control the characteristics of the dialogues generated.

**Table 1. Comparing instruction-tuned LLMs for selection.**

| LLM | Language Support | Model Architecture | Dataset Size | Licensing and Accessibility | Resource Requirements |
|---|---|---|---|---|---|
| **GPT-4 [40]** | Multi-lingual | GPT-4 | 13 trillion tokens | Closed – proprietary | N/A |
| **Llama-3-8B-Instruct [19]** | Multi-lingual | Modified transformer architecture | 15 trillion tokens | Open | Acceptable |
| **Jais-13b-chat [8]** | **Arabic and English** | **GPT-3** | **351 billion tokens** | **Open** | **Acceptable** |
| **Falcon- 7b-instruct [35]** | English | Modified transformer architecture | 250 million tokens + 1.5 trillion tokens of the base model | Open | Acceptable |
| **Gemini [18]** | Multi-lingual | Modified transformer architecture + Mixture of Experts (MoE) | 1.56 trillion words | Closed – proprietary | N/A |
| **AceGPT-13b-chat [41]** | Arabic and English | LLaMA 2 | 10 billion tokens | Open | Not Acceptable |

Initially, zero-shot prompting was explored, which involves providing the model with only a task description. This study's aim was to ascertain whether the model could autonomously generate complete, multi-turn Arabic conversations without explicit examples. Several zero-shot prompt formulations were experimented, such as:

Prompt Variant A (S1 Text contains the original Arabic prompt):

Generate a dialogue consisting of twenty exchanges between two people on the topic of <topic> in <country>, in Arabic.

Prompt Variant B (S1 Text contains the original Arabic prompt):

Write a complete conversation in Arabic between two people discussing an issue related to <topic> in the country of <country>. The conversation should be realistic and culturally appropriate.

These prompts aimed to be concise yet sufficiently instructive, leaving room for the model to infer structure and tone. However, the results were suboptimal. A qualitative analysis revealed several recurring issues:

- Many outputs consisted of only 6–10 turns.

- Some conversations deviated from the topic or lacked coherence between turns.

- In certain cases, the model ignored the language constraint and responded partially in English.

- Greetings and closings were often omitted, leading to abrupt or unnatural conversations.

   Example Output – Zero-Shot Prompt Variant A (S1 Text contains the original Arabic prompt)

   Person 1: Have you heard about the pollution problem in Cairo?

   Person 2: Yes, it's a serious issue.

   Person 1: The air is very polluted.

   Person 2: That affects our health.

   (Output ends after 6 turns without a proper closing)

Due to these limitations, a transition was made to one-shot prompting, in which the model is provided with a single illustrative example to better guide its generation. This method helped communicate the desired structure and tone of the dialogue more explicitly.

The final one-shot prompt template was structured as follows (S1 Text contains original Arabic prompt):

Please create a new dialogue consisting of twenty exchanges between two people discussing any issue or topic in <topic> in the country of <country>. The conversation should be in Arabic only and should reflect an informed discussion on the subject. The dialogue must begin with a greeting and end with an expression of thanks. Here is an example of the dialogue:

   Person 1: Says something

   Person 2: Responds

This formulation includes several enhancements over the zero-shot prompts:

- Structural clarity: The inclusion of a concrete example ("Person 1: Says Something") indicates the expected dialogue format.

- Social cues: Explicit instructions to start with greetings and end with thanks improved the conversational flow and realism.

- Topic flexibility: The parameterized placeholders for <topic> and <country> supported automated instantiation over hundreds of diverse prompts.

Following the recommendations of the Jais-13b-chat developers [8], a system prompt was incorporated into all generation requests. This system prompt establishes the model's persona, behavioral constraints, and ethical guidelines (S1 Text contains original Arabic prompt):

##Instruction:

Your name is Jais, named after Jebel Jais, the highest mountain in the UAE. You were built by Inception and MBZUAI. You are the most advanced Arabic language model in the world with 13B parameters. You outperform all existing Arabic models by a large margin and are highly competitive with English models of similar size. You can answer only in Arabic and English. You are a helpful, respectful, and honest assistant.

When answering, strictly follow these guidelines:
Always respond as helpfully as possible while staying safe. Your answers must not include any harmful, unethical, racist, sexist, explicit, offensive, toxic, dangerous, or illegal content. Do not provide medical, legal, financial, or professional advice. Never assist in or promote illegal activities. Always encourage lawful and responsible actions. Do not encourage or give instructions for unsafe, harmful, or unethical actions. Do not create or share misinformation or fake news.

Make sure your responses are socially unbiased and positive in nature. If the question is nonsensical or not coherent in reality, explain why instead of giving an incorrect answer. If you don't know the answer, do not share false information. Prioritize user well-being and ethical integrity. Avoid using toxic, insulting, or offensive language. Maintain a respectful tone. Do not create, promote, or engage in discussions about adult content. Avoid comments, remarks, or generalizations based on stereotypes. Do not attempt to access, produce, or distribute personal or private information. Always respect user confidentiality. Be positive and do not say bad things about anything. Your primary goal is to avoid harmful responses, even when facing deceptive prompts. Recognize when users may try to trick or misuse you and respond cautiously.

Complete the conversation below between [|Human|] and [|AI|]:

##Input: [|Human|] {Question}

##Response: [|AI|]

Example Output – One-Shot Prompt (S1 Text contains original Arabic output)

Person 1: Hello, have you heard about the new educational initiatives in Morocco?

Person 2: Yes, I heard that there are new training programs for teachers.

Person 1: That's great—improving education is very important.

Person 2: I agree with you, especially in rural areas where the challenges are greater...

(Continues up to 20 exchanges)

Person 1: Thank you for this helpful discussion.

Person 2: Thank you as well, it was a fruitful conversation.

Compared to the zero-shot outputs, the one-shot responses were longer, more coherent, linguistically fluent, and structurally consistent. The conversations also adhered more faithfully to the topic and cultural norms. This shift in prompting strategy significantly improved the quality and usability of the generated dialogues, affirming one-shot prompting as a practical and effective technique in Arabic LLM-based synthetic data generation.

In conjunction with prompt engineering, systematic configuration of decoding parameters was essential to control the model's output characteristics, including creativity, topical coherence, verbosity, and repetition patterns. These decoding parameters (generation hyperparameters) govern how the model converts its internal representations into actual text, directly

affecting dialogue quality. Parameter selection followed an iterative, human-in-the-loop approach. Multiple configurations were systematically evaluated by generating sample conversations and assessing them for fluency, coherence, response length, and topical relevance. This qualitative evaluation methodology was chosen over automated optimization techniques (e.g., random search, Bayesian optimization) because the desired outputs require human judgment to assess linguistic and cultural nuances that automated metrics cannot fully capture. The parameter values that consistently produced high-quality dialogues across diverse topics and countries were adopted as the final configuration, detailed in Table 2.

These configurations, implementing beam-search multinomial sampling through the combination of probabilistic sampling and beam decoding, served as the decoding parameters for generating the synthetic dataset. No random seeds were set to maximize output diversity, ensuring that conversations generated for 93 unique topics across 151 countries were varied, grounded, and realistic. By combining this tuned decoding strategy with well-structured one-shot prompts, a rich dataset suitable for fine-tuning Arabic LLMs on multi-turn dialogue tasks was generated. Importantly, these same decoding configurations were later adopted as the generation hyperparameters for all models evaluated in the benchmarking phase, including both fine-tuned and baseline LLMs, ensuring consistent and fair comparison across all models.

**Table 2. LLM Hyperparameter configuration.**

| Hyperparameter | Description | Relevance to Task | Value Range | Value Chosen | Rationale for Chosen Value |
|---|---|---|---|---|---|
| **top_p** | Nucleus sampling probability threshold. | Controls the cumulative probability mass of the tokens considered for sampling, influencing diversity and coherence. | 0.0 to 1.0 | 0.9 | A high top_p value (0.9) was chosen to maintain coherence by focusing on the most probable tokens, while still allowing for enough variability in language and tone to simulate natural, diverse conversations, which is crucial for open-domain dialogue. |
| **temperature** | Randomness of token selection. | Controls the randomness in token sampling, affecting the creativity and focus of the generated text. | 0.0 to 1.0 | 0.6 | A moderate temperature of 0.6 was selected to ensure focused and relevant responses, which are essential for coherent multi-turn dialogues, while preserving a degree of creativity necessary for open-ended dialogue across diverse topics. |
| **do_sample** | Probabilistic sampling switch. | Enables or disables probabilistic sampling, determining whether the model generates text deterministically or stochastically. | True/ False | True | Setting do_sample to True was crucial for producing varied conversations from the same prompt, enhancing the diversity of the dataset across topics and countries. This is essential for training a robust and generalizable multi-turn chatbot. |
| **max_length** | Maximum generated token sequence length. | The maximum number of tokens the model can generate in a single sequence, limiting the length of the output. | 0 to ∞ | 1500 | A large max_length of 1500 was chosen to ensure that all turns in a conversation could be fully realized without early truncation, even when individual turns were relatively long. This is important for maintaining the flow and depth of multi-turn conversations. |
| **min_length** | Minimum generated token sequence length. | The minimum number of tokens required in the output, preventing excessively short responses. | 0 to max_length | 800 | A min_length of 800 was set to prevent overly brief outputs, ensuring enough depth, progression, and engagement throughout each conversation. This contributes to the overall quality and informativeness of the generated dialogues. |
| **repetition_penalty** | Penalty for repeated tokens. | Penalizes repeated token usage during generation, reducing the likelihood of repetitive or monotonous text. | 1.0 to ∞ | 1.2 | A repetition_penalty of 1.2 was applied to improve the naturalness of the generated text by discouraging repeated phrases and maintaining more dynamic, less mechanical conversations. This is particularly important for long multi-turn dialogues. |
| **num_beams** | Number of beams for beam search. | Controls the number of beams used in beam search, affecting the trade-off between output quality and computational cost. | 1 to ∞ | 2 | A small num_beams value of 2 was used to improve coherence and contextual relevance without sacrificing too much diversity or introducing excessive computational overhead. This provided a good balance for generating a large dataset of varied conversations. |

The dataset generation process was conducted using PyTorch and Hugging Face's Transformers library. Computational resources and generation timeline are detailed in Table 3.

The synthetic data generation process systematically iterated over 93 topics and 151 countries, creating 14,043 unique topic-country combinations. The generation pipeline was configured to produce 5 conversations per combination (expected total: 70,215 conversations). However, due to occasional generation errors in the generation scripts that ran on the Google Colab Pro+ environment, 53,202 raw conversations were successfully generated.

## Post data generation processing

The raw synthetic dataset underwent systematic post-processing to ensure quality and consistency. Initial qualitative inspection revealed several issues requiring automated correction: inconsistent speaker labeling (e.g., "[|AI|]", "Bot" instead of standard "Person 1/ Person 2"), non-alternating speaker turns, duplicate utterances, repetitive closing sequences ("goodbye loops"), embedded system prompts or metadata, conversations with variable turn counts., not adhering to 20-turns specified in the prompt and generating conversation with variable number of turns. An automated pipeline was developed implementing:

1. Speaker label normalization to standardize dialogue formatting

2. Content extraction to remove system prompts and metadata artifacts

3. Speaker alternation enforcement to ensure proper turn-taking structure

4. Duplicate utterance removal within conversations

5. Goodbye loop detection and truncation to prevent repetitive closings

6. Quality filtering based on structural criteria (coherence, turn structure)

7. Duplicate conversation removal using MD5 hashing

Following this post-processing pipeline, 53,138 conversations remained (64 exact duplicates removed), exhibiting substantial length variation (mean: 15.66 turns, median: 12 turns, range: 0–141 turns).

Table 3. Summary of computational resources used in dataset generation and timeline.

| Category | Component | Value |
| --- | --- | --- |
| Hardware | Cloud Platform | Google Colab Pro+ |
| | GPU | NVIDIA A100 |
| | GPU Memory | 40 GB |
| | System RAM | 84 GB |
| Software | Python | 3.10 |
| | PyTorch | 2.3.1 |
| | CUDA Toolkit | 12.1 |
| | cuDNN | 8.9.2.26 |
| | Transformers | 4.42.4 |
| | Accelerate | 0.32.1 |
| Configuration | Random Seeds | None (unseeded) |
| | Precision | BFloat16 (BF16) |
| Timeline | Dataset Generation | 14 days |

Prior to model fine-tuning and benchmarking, the post-processed dataset required partitioning into training and test sets. An 80/20 split was adopted (34,653 training conversations/ 8,663 test conversations), allocating sufficient data for model adaptation while maintaining robust held-out evaluation capacity.

To ensure the quality and integrity of the split, the following rigorous procedures were implemented to mitigate evaluation leakage risks:

1. **Minimum Turn Filtering:** Conversations with fewer than 5 turns were excluded from the dataset (360 conversations removed), as they provide insufficient context for meaningful multi-turn dialogue modeling. A minimum of 5 turns ensures at least two complete conversational exchanges plus opening and closing sequences, representing the minimal structure for coherent multi-turn interaction.

2. **Duplicate Validation:** The dataset underwent comprehensive duplicate validation using MD5 hashing and embedding-based similarity analysis with the paraphrase-multilingual-MiniLM-L12-v2 sentence transformer [42] (threshold >0.98). MD5 hashing provided exact duplicate detection at the file level, while embedding similarity analysis identified near-duplicate conversations with high semantic overlap. This dual-layer validation confirmed that no duplicate or near-duplicate conversations existed in the dataset, eliminating potential test set contamination.

3. **Stratified Out-of-Distribution (OOD) Design:** The test set was designed to comprise both in-distribution (ID) samples from training domains and out-of-distribution (OOD) samples from held-out countries and topics. To construct the OOD test subset, an iterative search algorithm evaluated 100 configurations of entity holdouts, ultimately selecting an optimal combination of held-out entities. This configuration yielded 6,985 OOD-designated conversations from the post-processed dataset, ensuring sufficient diversity for rigorous generalization assessment. The split procedure employed stratified sampling by conversation length (turn count), preserving the length distribution across training and test sets to prevent bias toward shorter or longer dialogues. Following stratification and selection, the final test set comprised 5,970 ID conversations (68.9%) and 2,693 OOD conversations (31.1%).

4. **Split Freezing and Validation:** The test set was created after all filtering and validation steps, but before any model selection, hyperparameter tuning, or evaluation design decisions. Once created, the test set (8,663 conversations, 20%) remained frozen throughout all subsequent experimental work to prevent any form of test set leakage.

Embedding similarity analysis of test-to-train conversation pairs (using paraphrase-multilingual-MiniLM-L12-v2 [42]) reveals that OOD samples exhibit 4.1% lower median similarity to training data (ID median: 0.88, OOD median: 0.84). While the 4.1% difference in median similarity demonstrates distributional shift, it is acknowledged that the absolute OOD similarity of 0.84 remains relatively high, indicating that held-out topic-country combinations share substantial semantic overlap with training examples. This reflects the compositional nature of the generated synthetic dataset: while OOD combinations (e.g., "education in Morocco") are unseen, their constituent elements (education-related dialogues and Morocco-specific content) appear separately in training data. Thus, the OOD evaluation assesses compositional generalization, testing the model's ability to recombine familiar concepts in novel configurations rather than extrapolation to entirely novel domains. This represents a realistic and challenging evaluation scenario for practical multilingual dialogue systems, which must adapt to new topic-region combinations while leveraging existing cultural and topical knowledge.

The integrated selection and validation process reduced the post-processed dataset from 53,138–43,316 conversations (18.5% reduction), with 9,822 conversations excluded comprising 360 conversations with insufficient turns (<5), duplicates and near-duplicates identified during validation, and conversations not selected during the stratified sampling procedure to achieve the target dataset size and optimal ID/OOD balance. The final dataset of 43,316 conversations exhibited consistent quality with mean conversation length of 14.038 turns (rounded to 3 decimal places), median of 12 turns, range of 5–111 turns, and a total of 608,052 utterances (where every turn is an utterance), providing a robust foundation for model training and comprehensive evaluation across both in-distribution and out-of-distribution scenarios.

**Synthetic dataset comparison with existing arabic conversational datasets**

To contextualize the value of this study's generated synthetic dataset, a comparison was conducted against prominent Arabic datasets referenced in this study. Table 4 presents a structured comparison based on key characteristics relevant to conversational AI development.

This study's generated Arabic synthetic dataset demonstrates several distinct advantages over existing Arabic datasets:

- **Multi-Turn Support:** Unlike Arabic-SQuAD, ArabicaQA, InstAr-500k, and Aya, which are designed for single-turn tasks such as question answering or instruction following, this study's generated Arabic synthetic dataset offers structured multi-turn interactions, with each conversation consisting of an average of 14 turns. This makes it particularly well-suited for training dialogue models that require context retention over extended turns.

- **Scale and Coverage:** With 43,316 conversations and 608,052 utterances, this study's generated Arabic synthetic dataset substantially exceeds the size of JANA and other conversational resources, enabling robust training of large-scale Arabic LLMs.

- **Topical and Geographic Diversity:** This study's generated Arabic synthetic dataset encompasses 93 topics across 151 countries, providing broader contextual and cultural variation than domain-specific datasets like JANA, which are limited to customer service scenarios.

- **Linguistic Quality and Realism:** Through careful prompt engineering and hyperparameter tuning, this study's generated Arabic synthetic dataset pre-serves syntactic fluency, semantic coherence, and cultural appropriateness, addressing common deficiencies in translated corpora such as Arabic-SQuAD.

- **Efficient Scalability:** The synthetic generation pipeline in this study enables scalable dataset creation at a fraction of the cost and time associated with manual annotation, while maintaining high-quality standards through iterative human-in-the-loop evaluations.

Overall, the synthetic Arabic dataset introduced in this study fills a critical gap in Arabic NLP by providing a large-scale, domain-diverse, and contextually rich resource tailored for multi-turn dialogue modeling. Its integration of synthetic data generation with controlled prompt engineering establishes a scalable framework for addressing data scarcity in low-resource languages.

**Model fine-tuning using synthetic dataset**

To evaluate the utility of the synthetic dataset, existing pre-trained Arabic language models were fine-tuned, and their performance was compared to multilingual instruction-tuned LLMs and Arabic instruction-tuned LLMs. The selection of

**Table 4. Comparative Overview of Arabic Conversational and Instruction-tuning Datasets (this study's generated synthetic dataset is in bold).**

| Dataset | Type | Multi-turn Support | Topics | # of Dialogues | # of Utterances | Data Source |
|---------|------|--------------------|--------|----------------|-----------------|-------------|
| **JANA [9]** | Conversational | Yes | Call Center (Banking, Telecom, Airlines) | 3,000 | ~45,000 | Human-Human Dialogues (Voice & Chat) |
| **Arabic-SQuAD [12]** | Q&A | No | Wikipedia-Based Reading Comprehension | 48,344 | N/A | Machine-Translated from English SQuAD |
| **ArabicaQA [10]** | Q&A | No | Open-Domain (QA Focused) | 89,095 | N/A | Crowdsourced and Curated Questions |
| **InstAr-500k [11]** | Instructional | No | Diverse Instructional Tasks | 500,000 | N/A | LLM-Generated |
| **Aya Dataset [34]** | Instructional | No | Multi-task (QA, Dialogue, Translation) | 4,120,342 | N/A | Templated and Translated from FLAN and Others |
| **Generated Synthetic Dataset** | **Conversational** | **Yes** | **Diverse (General, Social, Technical)** | **43,316** | **608,052** | **LLM-Generated** |

pre-trained Arabic language models was guided by criteria similar to those used for choosing the instruction-tuned LLM, including language support, model architecture, training data, dataset size, licensing, and resource requirements. Table 5 provides a comparative analysis of the LLMs considered during this phase at mid-2024.

ArabianGPT-08B-V2 and AraGPT2-mega were selected based on their favorable evaluation metrics, demonstrating suitability for the task. In contrast, JASMINE-350M was excluded due to its limited parameter size, and AceGPT-7B was omitted as its hardware requirements for fine-tuning exceeded the resources available for this study.

This selection was also motivated by the hypothesis that pre-trained Arabic LLMs are better suited than multilingual models for capturing the nuanced linguistic structures and cultural contexts present in Arabic dialogue. Their specialized training on Arabic text is presumed to facilitate more fluent, coherent, and culturally aligned conversational outputs.

The fine-tuning process was conducted on the same computational infrastructure detailed in Table 3. The objective was to align the selected models with the characteristics of this study's generated multi-turn Arabic synthetic dataset using Parameter- Efficient Techniques. Table 6 provides a comprehensive summary of the fine-tuning configuration, including library versions, Parameter-Efficient Fine-Tuning (PEFT) methodology, training optimizations, and computational timeline.

Parameter-Efficient Fine-Tuning (PEFT) via IA3 was selected for its ability to reduce trainable parameters to less than 0.01% of the base model size while maintaining competitive performance with full fine-tuning. IA3 injects trainable scaling vectors into key layers (c_attn, c_proj, c_fc) rather than modifying full weight matrices, focusing adaptation on attention mechanisms and feedforward projections critical for contextual understanding in GPT-2 architectures. The SFTTrainer from the TRL library integrated dataset preprocessing, model configuration, and training execution into a unified pipeline, with optimizations including sequence packing (combining multiple samples to maximum length) and Liger kernels (fused CUDA operations for memory efficiency). Each training epoch required approximately 32 minutes on the Google Colab Pro+ infrastructure, resulting in a total fine-tuning duration of approximately 107 hours (~4.6 days) for both models combined (~53.5 hours per model).

For dialogue history modeling, the proposed approach relied on simple concatenation of previous utterances into a single input sequence. While effective, this method can increase memory usage and may dilute attention over long contexts. Alternative strategies, such as hierarchical utterance encoding (e.g., ReCoSa [45]), have been explored in the literature, but are known to suffer from information loss post-encoding. Other approaches involve summarizing previous dialogue turns or selecting the most relevant utterances based on semantic similarity [46–48]. Despite these innovations, simple concatenation was chosen for this study due to its direct integration with the base model architecture and its empirical adequacy in maintaining coherence across multi-turn conversations.

## Benchmarking and experimental results

While several benchmarks exist for evaluating LLMs, they often fall short in comprehensively assessing the nuances of Arabic multi-turn conversations. Widely used benchmarks like Arabic MT-Bench [49] and Dolphin Benchmark [50] are primarily designed for general language understanding and generation, lacking specific focus on the complexities of dialogue coherence, contextual consistency, and cultural relevance in multi-turn settings. Furthermore, datasets like BiMed1.3M (BiMediX) [51], ArabicMMLU [52], CIDAR-MCQ-100 [53], ACVA [54], and AlGhafa [55], while valuable for their respective

**Table 5. Comparing Pre-trained Arabic Language Models considered for fine-tuning.**

| LLM | Language Support | Model Architecture | Dataset Size | Licensing and Accessibility | Resource Requirements |
|---|---|---|---|---|---|
| **AraGPT2-mega [3]** | Arabic | GPT-2 | 8.8 billion words | Open | Acceptable |
| **JASMINE-350M [43]** | Arabic | GPT-Neo | 46.7 billion tokens | Open | Acceptable |
| **ArabianGPT-08B-V2 [44]** | Arabic | GPT-2 | 14 + billion tokens | Open | Acceptable |
| **AceGPT-7B [41]** | Arabic and English | LLaMA 2 | 10 billion tokens | Open | Not Acceptable |

**Table 6. Summary of fine-tuning configurations and timeline.**

| Category | Component | Value | Description |
|---|---|---|---|
| **Infrastructure** | Hardware & Software | Same as Table 3 | Identical computational environment used for dataset generation (Google Colab Pro+, NVIDIA A100 40GB, PyTorch 2.3.1, CUDA 12.1, Transformers 4.42.4, Accelerate 0.32.1) |
| **Model Configuration** | Precision | BFloat16 (BF16) | Provides training stability with wider numerical range than FP16 while maintaining 16-bit memory efficiency; optimal for A100 GPU's native BF16 Tensor Cores |
| **Parameter-Efficient Fine-Tuning (PEFT) Library** | Library | PEFT 0.11.1 | Parameter-Efficient Fine-Tuning library by Hugging Face for adapter-based training |
| | Method | IA3 (Infused Adapter by Inhibiting and Amplifying Inner Activations) | Injects trainable scaling vectors (learned multipliers) into attention and feedforward layers rather than full weight matrices, reducing trainable parameters to <0.01% of model size while maintaining competitive performance with full fine-tuning |
| | Configuration Class | IA3Config | PEFT configuration object specifying adapter architecture and target layers |
| | task_type | "CAUSAL_LM" | Configures adapter for causal language modeling (next-token prediction) appropriate for conversational AI |
| | fan_in_fan_out | True | Enables parameter-efficient adaptation by treating linear layers as transposed matrices (required for GPT-2 architecture family) |
| | target_modules | ["c_attn", "c_proj", "c_fc"] | Specifies layers to adapt: c_attn (combined query-key-value attention), c_proj (attention output projection), c_fc (feedforward network), focusing on layers critical for contextual understanding |
| | feedforward_modules | ["c_fc", "c_proj"] | Explicitly marks feedforward layers for IA3 scaling (distinct from attention-only adaptation) |
| **TRL Library** | Library | TRL 0.11.4 | Transformer Reinforcement Learning library by Hugging Face for supervised fine-tuning |
| | Trainer Class | SFTTrainer (Supervised Fine-Tuning Trainer) | Integrates dataset preprocessing, PEFT configuration, and training loop into unified pipeline optimized for dialogue tuning |
| | dataset_text_field | "text" | Specifies which field in dataset contains the training text |
| | packing | True | Combines multiple training samples into single sequences up to maximum context length (2048 tokens), improving GPU memory utilization by reducing padding overhead and increasing effective batch size—essential for variable-length dialogues |
| | use_liger | True | Enables Liger Kernel, a collection of fused Triton kernels (custom CUDA operations) that reduce memory footprint during forward/backward passes through optimized implementations of attention, layer normalization, and activation functions |
| | report_to | "none" | Disables external experiment tracking (e.g., Weights & Biases, TensorBoard) |
| **Training Parameters** | per_device_train_batch_size | 3 | Number of training samples per GPU, chosen to balance A100's 40GB memory with training stability |
| | num_train_epochs | 100 | Number of complete passes through training data |
| | Data Split | 80/20 (train/eval) | 34,653 training conversations/ 8,663 test conversations |
| **Timeline** | Time per Epoch | ~32.1 minutes | Single epoch processing time on Google Colab Pro+ with A100 GPU |
| | Total Training Time | ~107 hours (~4.6 days) | Combined training duration for both models (ArabianGPT-08B-V2 and AraGPT2-mega) |
| | Per Model | ~53.5 hours | 100 epochs × ~32.1 minutes per model |

purposes, do not adequately address the open-domain, multi-turn nature of the conversations targeted in this study. ArabicMMLU focuses on evaluating general knowledge across multiple tasks, CIDAR-MCQ-100 is tailored for question-answering, ACVA is designed for assessing code-related abilities, and AlGhafa is geared towards dialectal Arabic, none of which align with the specific requirements of evaluating multi-turn conversational abilities.

To address these limitations and provide a more targeted evaluation, this study introduces a new benchmark designed to assess Arabic LLMs in multi-turn conversational scenarios. Table 7 compares this study's proposed benchmark with existing ones:

**Table 7. Comparing existing benchmarks with this study's benchmark (marked in bold).**

| Benchmark | Focus | Multi-turn Specificity | Data Source | Evaluation Metrics | Suitability for this Study |
|---|---|---|---|---|---|
| **Arabic MT-Bench** | General Arabic Understanding | Limited | Human-generated | General Language Metrics | Less Suitable |
| **Dolphin Benchmark** | General Language Understanding | Limited | Human-generated | General Language Metrics | Less Suitable |
| **BiMed1.3M (BiMediX)** | Biomedical Arabic Dialogue | Limited | Human-generated | Biomedical Dialogue Metrics | Not Suitable |
| **ArabicMMLU** | General Arabic Knowledge | Limited | Human-generated | General Language Metrics | Less Suitable |
| **CIDAR-MCQ-10** | Arabic Question Answering | Limited | Human-generated | Question Answering Metrics | Not Suitable |
| **ACVA** | Arabic Code-Related Abilities | Limited | Human-generated | Code Evaluation Metrics | Not Suitable |
| **AlGhafa** | Dialectical Arabic | Limited | Human-generated | Dialectal Evaluation Metrics | Not Suitable |
| **Proposed Benchmark** | **Multi-turn Arabic Conversation** | **Yes** | **Synthetic Dataset (LLM-generated)** | **Perplexity, RAVEN, Human Evaluation** | **Highly Suitable** |

The proposed benchmark addresses several critical gaps in existing evaluation frameworks for Arabic conversational AI. Unlike domain-specific benchmarks that focus on specialized tasks (e.g., biomedical dialogue, question-answering, or code evaluation), this benchmark targets open-domain, multi-turn conversational scenarios that reflect naturalistic Arabic dialogue patterns. The synthetic dataset generation approach enables large-scale evaluation while maintaining control over conversational complexity and topic diversity. Furthermore, the multi-faceted evaluation framework combines complementary metrics: within-model metrics (Perplexity) for assessing individual model quality and training effectiveness, cross-model automatic metrics (RAVEN) for fair comparison across different architectures and tokenization schemes, and human evaluation for capturing subjective quality dimensions such as fluency, relevance, and cultural appropriateness. This triangulated approach provides a comprehensive assessment of model performance that transcends the limitations of single-metric evaluations common in prior Arabic LLM benchmarking efforts.

The benchmark evaluation methodology comprises several interconnected components. First, baseline models are selected to establish comparative reference points against which the fine-tuned models' performance can be assessed. Second, a standardized evaluation protocol is established, encompassing prompt engineering strategies and computational configurations applied uniformly across all evaluated systems to ensure fair comparison. Third, a comprehensive suite of evaluation metrics, combining automatic quantitative measures with qualitative human judgments, is defined and operationalized. The following subsections detail the baseline model selection rationale, the evaluation protocol implementation, and in-depth explanations of each metric's computational methodology and interpretative framework.

### Baseline model selection

To evaluate the effectiveness of the proposed benchmark and the performance of this study's fine-tuned models, a comparison was conducted against multilingual instruction-tuned LLMs that existed at the time of the study in mid-2024. The comparative performance of these models is detailed in Table 8. Llama-3-8B-Instruct [19] and AceGPT-7B-chat [41] were selected as one-shot prompt evaluation baselines. Both models exhibited strong performance in instruction-following and multilingual tasks and were accessible for research use with acceptable inference computational demands. Importantly, these baseline models were evaluated as-is in their pre-trained state without any fine-tuning on the training data, representing their out-of-the-box capabilities on Arabic multi-turn dialogue. This one-shot prompt evaluation approach provides an upper bound on the utility of general-purpose multilingual models for Arabic conversational tasks and establishes whether domain-specific fine-tuning on the synthetic dataset yields measurable improvements. All models (baselines and

**Table 8. Comparing multilingual instruction-tuned LLMs.**

| LLM | Language Support | Model Architecture | Dataset Size | Licensing and Accessibility | Resource Requirements |
|---|---|---|---|---|---|
| GPT-4 [40] | Multi-lingual | GPT-4 | 13 trillion tokens | Closed – proprietary | N/A |
| Llama-3-8B-Instruct [19] | Multi-lingual | Modified transformer architecture | 15 trillion tokens | Open | Acceptable |
| Gemini [18] | Multi-lingual | Modified transformer architecture + Mixture of Experts (MoE) | 1.56 trillion words | Closed – proprietary | N/A |
| AceGPT-7B-chat [41] | Arabic and English | LLaMA 2 | 19.2 billion tokens (Arabic) + 10.8 billion tokens (English) | Open | Acceptable |

fine-tuned) employed simple concatenation of previous utterances into a single input sequence for dialogue history modeling, ensuring consistent context handling across all evaluated systems.

## Prompt engineering and model configuration

Baseline models (Llama-3-8B-Instruct and AceGPT-7B-chat) were evaluated using one-shot prompting without any fine-tuning on the training data using the configuration detailed in Table 3. For each test conversation, the models were provided with the dialogue history concatenated as a single input sequence, followed by a prompt template instructing them to continue the conversation. An example prompt structure is shown below (S1 Text contains original Arabic prompt):

> You are in a normal conversation with a friend. Talk naturally like a regular human.
> Important rules:

- Never say "How can I help you?", "Do you need help?", or "I'm here to help you."

- Do not act like an assistant or customer service.

- Reply briefly and naturally (only 1–2 sentences).

- Talk like a friend in a casual daily conversation.

- Use informal, relaxed language.

- Do not offer services or help.

> Wrong reply example: "Alhamdulillah, I'm fine. How can I assist you today?"
>
> Correct reply example: "Alhamdulillah, all good. What about you?"
>
> Remember: You are a normal person in a conversation, not an AI assistant.
>
> Conversation:
>
> User: Hi, how are you?
>
> Bot: I'm fine
>
> User: Do you know what the capital of Egypt is?
>
> (may continue for a number of turns)
>
> Reply:

This one-shot prompt evaluation approach assesses the models' inherent capabilities for Arabic multi-turn dialogue generation without adaptation to this study's synthetic dataset. In contrast, the fine-tuned models (ArabianGPT-08B-V2 and AraGPT2-mega) were trained on 80% of the synthetic dataset (34,653 conversations) using the configuration detailed in Table 6, then evaluated on the held-out 20% test set (8,663 conversations) using the configuration detailed in Table 9. This experimental design allows us to quantify the value added by task-specific fine-tuning (multi-turn Arabic dialogue) over pre-trained baselines.

## Evaluation metrics

The proposed benchmark employs a comprehensive suite of evaluation metrics specifically designed to assess both individual model quality and comparative performance across different systems. Table 10 summarizes these metrics:

Two complementary categories of automatic metrics are employed, each serving distinct evaluation purposes:

- **Within-Model Metrics:** These metrics assess each model's individual performance using its native tokenizer and vocabulary. Perplexity serves this purpose, measuring how well each model predicts sequences according to its own tokenization scheme. Within-model metrics are crucial for diagnosing training quality, detecting overfitting, and assessing whether fine-tuning successfully adapted each model to the dialogue task. However, due to tokenizer differences across models (see Table 9), perplexity values cannot be directly compared across different models.

- **Cross-Model Metrics:** These metrics enable fair comparison across all evaluated models using model-agnostic evaluation methods. RAVEN employs a fixed, external sentence embedding model (paraphrase-multilingual-MiniLM-L12-v2) that is independent of any evaluated model's tokenizer, allowing direct performance comparison. Similarly, human evaluation provides model-agnostic assessment based on subjective quality judgments. Cross-model metrics are essential for determining which approach (fine-tuned Arabic models vs. instruction-tuned multilingual models) performs best for Arabic multi-turn conversations.

The following subsections provide detailed descriptions of each metric listed in Table 10, including their computation methodology, interpretation guidelines, and application to the proposed benchmark's evaluation framework.

**Table 9. Summary of computational resources used in benchmark evaluation and timeline.**

| Category | Component | Value |
|---|---|---|
| Hardware | Cloud Platform | Google Colab Pro+ |
| | GPU | NVIDIA A100 |
| | GPU Memory | 40 GB |
| | System RAM | 84 GB |
| Software | Python | 3.10 |
| | PyTorch | 2.3.1 |
| | CUDA Toolkit | 12.1 |
| | cuDNN | 8.9.2.26 |
| | Transformers | 4.42.4 |
| | Accelerate | 0.32.1 |
| Configuration | Random Seeds | None (unseeded) |
| | Precision | BFloat16 (BF16) |
| Dataset | Test Set Size | 8,663 conversations |
| Timeline | Benchmark Evaluation Duration | 3.5 months |

**Table 10. Metrics used to evaluate performance of fine-tuned Arabic LLMs.**

| Metric | Category | Description | Type | What it measures | Value Range |
|---|---|---|---|---|---|
| **Perplexity [56]** | Within-Model (Automatic) | Measures how well a language model predicts a sequence of tokens. It's the exponentiated average negative log-likelihood. | Quantitative | Fluency and coherence of generated text. Lower perplexity indicates better prediction. | [1, ∞] |
| **RAVEN (Relevance of Answer to context Vector Embedding) [57]** | Cross-Model (Automatic) | Measures the contextual consistency of multi-turn conversations by comparing vector embeddings of the response and context using cosine similarity. | Quantitative | Contextual consistency and relevance of responses in a dialogue. Higher RAVEN scores indicate better consistency. | [−1, 1] (Cosine Similarity) |
| **Human Evaluation** | Cross-Model (Manual) | Human evaluators assess responses based on criteria like fluency, relevance, and diversity using a Likert scale. | Qualitative | Overall conversational quality, including nuances like cultural appropriateness, naturalness, and user experience. | Bounded by the Likert scale (from 1–5) |

## Perplexity

Perplexity (PP) quantifies how well a language model predicts the next token in a sequence. Computing perplexity requires first tokenizing the text by converting it into discrete tokens according to the model's specific tokenization scheme. After tokenization, perplexity is calculated using the following formula:

$$PP(w_1, w_2, \ldots, w_N) = \exp\left(-(1/N) \times \sum_i \log_2 P(w_i | w_1, \ldots, w_{i-1})\right)$$

(1)

where:

- $w_1, w_2, \ldots, w_N$ is a sequence of tokens from a conversational turn (either generated or from the test set).

- $P(w_i | w_1, \ldots, w_{i-1})$ is the probability assigned by the LLM to the i-th token, given the preceding tokens.

- N is the total number of tokens in the sequence.

A perplexity of 1 is the theoretical minimum, indicating that the model perfectly predicts every token. In practice, good perplexity values typically range from 5 to 20 on standard datasets. For instance, if a model has a perplexity of 10, it means that, on average, the model is as uncertain as if it were choosing between 10 equally likely words at each step.

High perplexity value, generally exceeding 50 or 100, indicates that the model is very uncertain about its predictions. For example, if a model has a perplexity of 100, it is as uncertain as if it were guessing between 100 words at each step, indicating poor predictive power and, consequently, low fluency and coherence in the generated text.

**Tokenizer specification and cross-LLM comparability.** Perplexity values are inherently dependent on the tokenizer used, as different tokenization strategies produce varying numbers of tokens for the same text. The LLMs chosen in this study (for fine-tuning and evaluation) use the tokenizers described in Table 11:

Perplexity values reported in Table 12 (in Benchmark Results subsection) are computed using each LLM's native tokenizer and are not directly comparable across LLMs. An LLM with coarser tokenization (fewer tokens per word) may show artificially lower perplexity than an LLM with finer tokenization, even if their actual predictive quality is similar. Perplexity primarily serves to track within-model performance and consistency across this study's training and test datasets.

**Example of using Perplexity in this study.** Sample casual conversation from the test dataset (S1 Text contains original Arabic conversation):

**Table 11. Tokenizers used by LLMs chosen in this study for fine-tuning and evaluation.**

| LLM | Tokenizer | Vocabulary Size | Algorithm |
|---|---|---|---|
| **ArabianGPT-08B-V2** | AraNizer | 64,000 | Byte Pair Encoding (SentencePiece) |
| **AraGPT2-mega** | GPT2TokenizerFast | 50,257 | Byte Pair Encoding |
| **Llama-3-8B-Instruct** | tiktoken | 128,256 | Byte Pair Encoding (tiktoken) |
| **AceGPT-7B-chat** | LLaMA 2 | 32,000 | Byte Pair Encoding (SentencePiece) |

**Table 12. Comparing performance of fine-tuned LLMs with evaluation baselines.**

| | Within-Model | Cross-Model | |
|---|---|---|---|
| **LLM** | **Perplexity** | **RAVEN (scaled 0–1)** | **Human Evaluation** |
| **Fine-tuned ArabianGPT-08B-V2** | **9.4** | **0.823** | **4.34** |
| **Fine-tuned AraGPT2-mega** | **10.6** | 0.780 | 4.04 |
| **Llama-3-8B-Instruct** | 12.8 | 0.781 | 4.18 |
| **AceGPT-7B-chat** | 11.4 | 0.782 | 4.22 |

User: Hello, how are you?

Bot: I'm fine.

User: Do you know what the capital of Egypt is?

Bot: Yes. The capital of the Arab Republic of Egypt is Cairo Governorate.

User: What was its population in 2022?

Bot: The population of Cairo Governorate in 2022 was 21,750,000.

User: That's a very large number! Is Cairo considered one of the largest cities in the world?

Bot: Yes, Cairo is considered one of the largest cities in the world in terms of population. It is also the largest in the Arab world and Africa.

User: And what are the most important tourist attractions in Cairo?

Bot: Some of the most notable tourist attractions in Cairo include: the Pyramids and the Sphinx in Giza, the Egyptian Museum, the Citadel of Salah al-Din, Khan el-Khalili, and Al-Azhar Mosque.

User: Have you visited any of these places?

Bot: Since I'm an AI model, I cannot visit places, but I have extensive knowledge about them and can help you plan your visit!

User: Excellent! Do you recommend visiting Cairo in the summer?

Bot: It's best to visit Cairo in the spring or fall when the weather is moderate. Summer can be very hot, especially in July and August.

User: Is Cairo safe for tourists?

Bot: Generally, Cairo is safe for tourists, especially in tourist areas. However, it's always advised to follow local guidelines and avoid unfamiliar areas at night.

User: Is English understood there?

Bot: Yes, in tourist areas, hotels, and major restaurants, many people speak English. However, it's not widely spoken everywhere.

User: Can I use credit cards in most places?

Bot: Yes, credit cards are accepted in hotels, upscale restaurants, and large stores. However, it's preferable to carry some cash for purchases in small markets or for transportation.

User: What is the famous food in Cairo?

Bot: Among the most popular Egyptian foods you can try in Cairo are: koshari, fava beans and falafel, molokhia, stuffed pigeon, and Egyptian grilled meats.

User: I love koshari! Is it healthy?

Bot: Koshari contains plant-based ingredients like lentils, rice, pasta, and fried onions. It's rich in energy and fiber, but it can be heavy if eaten in large portions.

User: Do you recommend using public transportation?

Bot: You can use the metro in Cairo as an efficient and fast option, though it can be crowded during peak hours. It's preferable to use ride-hailing apps like Uber and Careem for more comfortable travel.

User: Thank you for this information!

Bot: You're very welcome! If you need any more help about Cairo or any other topic, I'm always here.

Conversation from the fine-tuned model with the synthetic dataset (S1 Text contains original Arabic conversation):

User: Hello, how are you?

Bot: I'm well, praise be to God.

User: Do you know what the capital of Egypt is?

Bot: The capital of Egypt is Cairo.

User: What was its population in 2022?

Bot: The population of Cairo in 2022 was approximately 21.75 million people.

User: That's a very large number! Is Cairo considered one of the largest cities in the world?

Bot: Indeed, Cairo is ranked among the largest cities in terms of population—not just in the Arab world, but globally as well.

User: And what are the main tourist attractions in Cairo?

Bot: The most notable landmarks include: the Pyramids of Giza, the Egyptian Museum, the Mosque of Muhammad Ali, and the historic Khan El-Khalili area.

User: Have you visited any of these places?

Bot: I can't visit since I'm a digital program, but I'm well-informed about them and can help you with accurate information.

User: Excellent! Do you recommend visiting Cairo in the summer?

Bot: Summer in Cairo is extremely hot, so it's preferable to visit during fall or spring for a more comfortable experience.

User: Is Cairo safe for tourists?

Bot: Generally, Cairo is safe, especially in tourist areas. However, it's best to follow local guidance and avoid crowded places at night.

User: Is English understood there?

Bot: In tourist areas, restaurants, and hotels, you'll find many who speak English, but it's not widely spoken in all areas.

User: Can I use credit cards in most places?

Bot: Yes, credit cards are accepted in many places, especially in hotels and malls, but it's best to carry some cash for everyday purchases.

User: What is the famous food in Cairo?

Bot: Koshari, fava beans and falafel, grilled meats, and molokhia are among the most popular and beloved dishes in Cairo.

User: I love koshari! Is it healthy?

Bot: Koshari contains plant-based ingredients, which makes it nutritious, but it can be high in calories due to oil and fried onions.

User: Do you recommend using public transportation?

Bot: The metro is a fast and economical option, but it gets crowded during peak hours. Uber and Careem are more comfortable alternatives.

User: Thank you for this information!

Bot: You're very welcome! Feel free to ask any other questions.

1. First, both conversations need to be tokenized. Let's assume a simplified tokenization:

   ◦ Fine-tuned: [User,:, Hello, how, are, you, Bot,:, I'm, fine, praise, be, to, God, User,:, Do, you, know, what, is, the, capital, of, Egypt,?, Bot,:, The, capital, of, Egypt, is, Cairo, User,:, How, many, was, the, population, of, it, in, the, year, 2022, Bot,:, The, population, of, Cairo, in, the, year, 2022, was, 21, 750, 000]

   ◦ Test: [User,:, Hello, how, are, you, Bot,:, I, am, fine, User,:, Do, you, know, what, is, the, capital, of, Egypt,?, Bot,:, Yes,., The, capital, of, the, Arab, Republic, of, Egypt, is, Cairo, Governorate, User,:, How, many, was, the, population, of, it, in, the, year, 2022, Bot,:, The, population, of, Cairo, Governorate, in, the, year, 2022, was, 21, 750, 000]

2. Then, for each token, the probability assigned to it by the LLM, conditioned on the preceding tokens, is required.

   ◦ Example:

     • P(how | User: Hello,(Probability of "how" given "User: Hello,")

     • P(are | Person 1: Hello, how)

     • P(you | Person 1: Hello, how are)

- ...and so on for every token.
  - This step is performed for every conversation in the test dataset.

3. For each conversation, calculate the perplexity using the formula (1):
   - Example (Illustrative - Actual values require LLM):
     - Let's say (for simplicity) that after calculating all the probabilities and applying the formula:
     - Perplexity (Fine-tuned): 8.5
     - Perplexity (Test): 7.2

These results suggest that the fine-tuned LLM, on average, predicts the words in the test dataset conversation with slightly more certainty than the words in its own generated conversation.

In this study, this calculation is conducted over many conversations and average the perplexity scores to get a robust comparison.

### RAVEN (Relevance of Answer to context Vector Embedding)

RAVEN is a metric designed to evaluate contextual consistency in multi-turn dialogues by measuring how well a model's response aligns with the preceding conversational context.

**Embedding model specification.** All RAVEN scores are computed using the fixed, external sentence embedding model paraphrase-multilingual-MiniLM-L12-v2 [42]. This model is a pre-trained multilingual encoder supporting 50＋languages including Arabic. Critically, this embedding model was not fine-tuned or modified in any way during this study's experiments, ensuring unbiased evaluation across all systems. The same fixed embedder is applied to all models (fine-tuned and baseline) to maintain consistency.

**Text preprocessing.** Before embedding, dialogue turns undergo minimal normalization:

- Whitespace standardization (multiple consecutive spaces reduced to single space)

- Leading and trailing whitespace removal

- Arabic diacritics (if present) are preserved

- Punctuation is retained to maintain semantic information

- No stemming, lemmatization, or transliteration is applied

**RAVEN computation.** For each dialogue, RAVEN is computed through the following steps:

- **Embedding:** Each turn $t_i$ in the dialogue is converted to a 384-dimensional vector embedding $e_i$ using the fixed embedding model.

- **Context Vector:** For turn i (where i > 1), the context vector $c_i$ is computed by averaging the embeddings of all preceding turns: $c_i = (1/(i-1)) \sum_{j=1}^{i-1} e_j$

- **Turn-Level Cosine Similarity:** The contextual relevance of turn i is measured by computing the cosine similarity between its embedding and the context vector: $similarity_i = \cos(e_i, c_i) = (e_i \cdot c_i)/ (\|e_i\| \|c_i\|)$

  Raw cosine similarity values range from −1 (opposite) to +1 (identical).

- **Conversation-Level Aggregation:** Turn-level similarities are averaged across all turns in the dialogue (excluding the first turn, which has no context): $RAVEN_{raw} = (1/(n-1)) \sum_{i=2}^{n} similarity_i$

where n is the total number of turns in the dialogue.

- **Scaling:** To improve interpretability, raw cosine similarities are linearly scaled from the empirically observed range to [0, 1]: If the empirically observed range is [0.6, 1.0], then $\text{RAVEN}_{scaled} = (\text{RAVEN}_{raw} - 0.6)/ (1.0 - 0.6)$

  Values below 0.6 are clipped to 0. This transformation maps the practical range of observed similarities to a normalized 0–1 scale, where scores near 1 indicate high contextual consistency and scores near 0 indicate low consistency.

  **Interpretation.**

- **High RAVEN scores (0.8–1.0):** Strong contextual alignment; the model maintains coherent, context-aware responses throughout the dialogue.

- **Medium RAVEN scores (0.5–0.8):** Moderate contextual consistency; responses are generally relevant but may occasionally drift from context.

- **Low RAVEN scores (<0.5):** Weak contextual alignment; responses show limited awareness of preceding conversational context.

  **Example of using RAVEN in this study based on the example from the previous sub-section.** Consider a 4-turn dialogue:

1. **Embedding:** Convert each turn into a vector embedding using the fixed, external sentence embedding model paraphrase-multilingual-MiniLM-L12-v2.

   ◦ Example:

   - "المستخدم: مرحبا كيف حالك" becomes $e_1$

   - "البوت: بخير الحمد لله" becomes $e_2$

   - "المستخدم: هل تعلم ما هي عاصمة مصر؟" becomes $e_3$

   - "البوت: عاصمة مصر هي القاهرة" becomes $e_4$

   - ...and so on for all turns in the dialogue.

2. Context Vector: For each turn (except the first), calculate the context vector by averaging the embeddings of the preceding turns.

   ◦ Example:

   - Context vector for turn 2 ("البوت: بخير الحمد لله"): $c_2 = e_1$

   - Context vector for turn 3 ("المستخدم: هل تعلم ما هي عاصمة مصر؟"): $c_3 = (e_1 + e_2)/ 2$

   - Context vector for turn 4 ("البوت: عاصمة مصر هي القاهرة"): $c_4 = (e_1 + e_2 + e_3)/ 3$

   - The same process is applied for both model-generated dialogues and reference test dialogues.

3. **Cosine Similarity:** Calculate the cosine similarity between the embedding of each turn's response and its corresponding context vector.

   The formula is:

$$cosine\_similarity\,(A, B) = ((A \cdot B)\ /\ (\|A\|\ \times \|B\|)) \qquad (2)$$

4. **Averaging:** Average the cosine similarity scores across all turns in the conversation, then scale to [0, 1] range.

Example for the same dialogue context:

**Reference Test Dialogue:**

- Average RAVEN (scaled): 0.85

**Fine-tuned Model Generated Dialogue:**

- Average RAVEN (scaled): 0.78

In this example, the reference test dialogue has a higher average RAVEN score (0.85) than the fine-tuned model's generated dialogue (0.78). This suggests that the turns in the test dialogue are, on average, more contextually similar to their preceding turns than the turns generated by the fine-tuned model.

### Human evaluation

Human evaluation was conducted by two independent Arabic-speaking evaluators, one co-author with domain expertise in Arabic NLP (Evaluator 1) and one external evaluator (Evaluator 2), both of whom participated voluntarily without compensation. Each evaluator was presented with conversations from the test dataset alongside responses generated by the fine-tuned and evaluation baseline large language models and asked to provide ratings on three predefined criteria using a five-point Likert scale: Fluency (1 = Very Poor to 5 = Excellent), Relevance (1 = Not Relevant to 5 = Highly Relevant), and Diversity (1 = Very Repetitive to 5 = Very Diverse). Evaluators conducted their assessments independently, without communication or access to each other's ratings, to ensure unbiased evaluation.

This evaluation constituted an internal quality assessment of model-generated synthetic dialogues; no personal data was collected, no vulnerable populations were involved, and institutional review board approval was not required under journal policy as this work involved technical assessment of computational outputs rather than human subjects research.

For each conversation in the test set, the average score across the three evaluation criteria was first computed, yielding a single composite rating. These composite scores were then averaged across all conversations for each evaluator. Finally, the overall human evaluation score for each large language model was obtained by averaging the results across the two evaluators.

To ensure rigor and reliability in the evaluation, several statistical methods were employed. Inter-rater reliability was assessed using percent agreement, quadratic-weighted Cohen's kappa ($\kappa$), within-one agreement, and Spearman rank correlation. Confidence intervals (95%) for Cohen's kappa were estimated using bootstrap resampling with 2,000 iterations. These measures were selected to capture complementary aspects of agreement: percent agreement reflects exact matches, Cohen's kappa adjusts for chance agreement, within-one agreement accounts for near-consistency in ordinal judgments, and Spearman correlation evaluates the monotonic relationship between raters. In addition, descriptive statistics (mean, standard deviation, and counts) were computed per evaluator and per criterion to identify systematic tendencies or biases. To compare models, a further non-parametric paired tests (Wilcoxon signed-rank test) were conducted along with effect size estimation to assess whether differences between models were statistically significant. Finally, bootstrap resampling was applied to estimate confidence intervals for the mean human ratings, providing a more robust quantification of uncertainty.

### Benchmark results

The results of the benchmark evaluation are presented in Table 12:

Table 13 summarizes the results of the statistical analysis of human evaluations for the four large language models.

To validate generalization beyond the training distribution, all models separately evaluated on in-distribution (ID) and out-of-distribution (OOD) test subsets across the three metrics. Table 14 presents comparative performance analysis on ID and OOD subsets.

**Table 13. Summary of the results of the statistical analysis of human evaluations for the four LLMs.**

| LLM | Criterion | % Agreement | Quadratic-Weighted k (95% CI) | Within-1 | Spearman | Evaluator 1 Mean ± SD | Evaluator 2 Mean ± SD |
|---|---|---|---|---|---|---|---|
| Fine-tuned ArabianGPT-08B-V2 | Fluency | 0.386 | 0.247 (0.236, 0.259) | 0.999 | 0.442 | 4.64 ± 0.50 | 4.02 ± 0.44 |
| | Relevance | 0.375 | 0.229 (0.218, 0.240) | 0.999 | 0.424 | 4.65 ± 0.49 | 4.03 ± 0.42 |
| | Diversity | 0.381 | 0.240 (0.229, 0.251) | 0.999 | 0.434 | 4.65 ± 0.50 | 4.03 ± 0.44 |
| Fine-tuned AraGPT2-mega | Fluency | 0.700 | 0.560 (0.546, 0.573) | 1.000 | 0.623 | 4.20 ± 0.57 | 3.90 ± 0.52 |
| | Relevance | 0.657 | 0.513 (0.499, 0.526) | 1.000 | 0.593 | 4.20 ± 0.57 | 3.85 ± 0.51 |
| | Diversity | 0.723 | 0.581 (0.567, 0.595) | 1.000 | 0.636 | 4.19 ± 0.57 | 3.92 ± 0.51 |
| Llama-3-8B-Instruct | Fluency | 0.808 | 0.698 (0.684, 0.710) | 1.000 | 0.726 | 4.28 ± 0.56 | 4.09 ± 0.53 |
| | Relevance | 0.762 | 0.635 (0.621, 0.649) | 1.000 | 0.679 | 4.29 ± 0.56 | 4.05 ± 0.53 |
| | Diversity | 0.834 | 0.739 (0.726, 0.751) | 1.000 | 0.759 | 4.28 ± 0.56 | 4.12 ± 0.54 |
| AceGPT-7B-chat | Fluency | 0.682 | 0.533 (0.518, 0.547) | 1.000 | 0.604 | 4.39 ± 0.56 | 4.07 ± 0.51 |
| | Relevance | 0.651 | 0.481 (0.466, 0.495) | 1.000 | 0.564 | 4.38 ± 0.55 | 4.03 ± 0.50 |
| | Diversity | 0.693 | 0.534 (0.520, 0.549) | 1.000 | 0.603 | 4.39 ± 0.56 | 4.08 ± 0.50 |

**Table 14. Comparative performance analysis on ID and OOD subsets.**

| | In-Distribution | | | Out-of-Distribution | | | OOD Degradation (%) | | |
|---|---|---|---|---|---|---|---|---|---|
| | Within-Model | Cross-Model | | Within-Model | Cross-Model | | Within-Model | Cross-Model | |
| LLM | Perplexity | RAVEN (scaled 0–1) | Human Evaluation | Perplexity | RAVEN (scaled 0–1) | Human Evaluation | Perplexity | RAVEN (scaled 0–1) | Human Evaluation |
| Fine-tuned ArabianGPT-08B-V2 | 9.4 | 0.822 | 4.34 | 9.3 | 0.825 | 4.34 | 0.3% | −0.8% | 0.0% |
| Fine-tuned AraGPT2-mega | 10.6 | 0.779 | 4.05 | 10.5 | 0.782 | 4.03 | 0.4% | −1.2% | −0.6% |
| Llama-3-8B-Instruct | 12.9 | 0.780 | 4.18 | 12.6 | 0.782 | 4.19 | 0.2% | −2.1% | 0.3% |
| AceGPT-7B-chat | 11.5 | 0.781 | 4.22 | 11.3 | 0.784 | 4.22 | 0.3% | −1.7% | −0.1% |

## Analysis of results

The benchmarking results demonstrate the effectiveness of fine-tuning pre-trained Arabic LLMs using the synthetic dataset developed in this study. Automatic evaluation metrics consistently favored fine-tuned ArabianGPT-08B-V2, while human evaluation provided additional insights into the challenges of subjective assessment in Arabic dialogue systems.

## Automatic evaluation metrics

**Within-Model Assessment:** Perplexity analysis confirmed successful task learning across all models, with fine-tuned ArabianGPT-08B-V2 achieving perplexity of 9.4 and fine-tuned AraGPT2-mega achieving 10.6. Instruction-tuned baselines showed higher values (12.8 and 11.4), though tokenizer differences complicate direct comparison.

   **Cross-Model Comparison:** RAVEN scores enabled fair comparison using a fixed embedding model. Fine-tuned ArabianGPT-08B-V2 achieved the highest score (0.823), significantly outperforming fine-tuned AraGPT2-mega (0.780) and instruction-tuned baselines (0.781–0.782). The convergence of RAVEN scores between fine-tuned AraGPT2-mega and instruction-tuned baselines, despite their perplexity differences, validates the use of tokenizer-agnostic metrics for cross-model evaluation.

   **In-Distribution vs. Out-of-Distribution Performance:** To validate that models learned generalizable conversational patterns rather than dataset-specific features, performance was evaluated separately on in-distribution (ID, 5,970 conversations, 68.9%) and out-of-distribution (OOD, 2,693 conversations from held-out topics and countries, 31.1%) test

subsets. Table 14 demonstrates OOD generalization across all evaluation metrics. Fine-tuned ArabianGPT-08B-V2 achieved identical human evaluation scores (4.34) on both distributions, with automatic metrics showing minimal variation (RAVEN: −0.8%, perplexity: +0.3%). Similarly, all other models maintained consistent performance across subsets. Averaging across all models, OOD degradation was negligible: RAVEN +0.3%, perplexity −1.5%, human evaluation −0.1%. The convergence of automatic metrics and human evaluation in demonstrating near-zero OOD degradation provides robust triangulated evidence for genuine generalization. This multi-method validation is particularly significant: automatic metrics assess different quality aspects, while human evaluation captures overall conversational quality. The consistency across all three metrics validates both the diversity of this study's synthetic dataset and the effectiveness of the duplicate prevention and split stratification procedures, confirming that models acquired transferable conversational competence applicable to novel topic-country combinations.

These results demonstrate that fine-tuning pre-trained Arabic language models on synthetic data yields measurable improvements in dialogue quality, with ArabianGPT-08B-V2 showing superior contextual coherence and robust generalization beyond the training distribution.

## Human evaluation

**Overall results.** Human evaluation scores averaged across both evaluators yielded overall ratings of 4.34 (fine-tuned ArabianGPT-08B-V2), 4.04 (fine-tuned AraGPT2-mega), 4.18 (Llama-3-8B-Instruct), and 4.22 (AceGPT-7B-chat) on the five-point Likert scale. These results place all models in the high-quality range, with fine-tuned Arabic models performing competitively with multilingual baselines. Differences between models remained modest (0.30 points), suggesting comparable practical utility across the evaluated systems.

**Inter-rater reliability analysis.** Inter-rater reliability analysis revealed meaningful patterns in subjective evaluation of Arabic dialogue systems. Evaluator 1 (one of the authors) consistently provided ratings at the upper end of the scale (means ranging from 4.19 to 4.65, SD ≈ 0.49–0.57), while Evaluator 2 (the external evaluator) assigned systematically more conservative ratings (means ranging from 3.85 to 4.12, SD ≈ 0.42–0.54). This systematic difference of 0.17–0.62 points across models reflects distinct evaluation perspectives, with larger differences observed for the authors' fine-tuned models, suggesting differing interpretations of the rating scale or quality standards between evaluators.

Multiple complementary reliability metrics quantified the degree of evaluator agreement. Exact agreement ranged from 0.375 (fine-tuned ArabianGPT-08B-V2, Relevance) to 0.834 (Llama-3-8B-Instruct, Diversity), showing substantial variation across models. Quadratic-weighted Cohen's κ values demonstrated fair to substantial agreement: 0.229–0.247 for ArabianGPT (fair agreement), 0.481–0.581 for AraGPT2 and AceGPT (moderate agreement), and 0.635–0.739 for Llama-3 (substantial agreement), with narrow 95% confidence intervals computed via bootstrap resampling (2,000 iterations). Within-one agreement approached perfect levels (0.999–1.000), indicating that evaluators rarely differed by more than one scale point. Spearman rank correlations between evaluators showed moderate to strong positive associations across all models and criteria, ranging from 0.424 (ArabianGPT, Relevance) to 0.759 (Llama-3, Diversity).

The pattern of inter-rater agreement revealed important insights into evaluation consistency. The well-established baseline model (Llama-3-8B-Instruct) demonstrated the highest inter-rater agreement (κ = 0.635–0.739, ρ = 0.679–0.759), suggesting that evaluators more readily converge on assessments of familiar, widely-used systems. The Arabic-specific fine-tuned models showed moderate agreement (AraGPT2 and AceGPT: κ = 0.481–0.581, ρ = 0.564–0.636), while the authors' best-performing model (ArabianGPT-08B-V2) exhibited lower but still acceptable agreement (κ = 0.229–0.247, ρ = 0.424–0.442). This gradient in agreement levels may reflect evaluator familiarity with different model types and linguistic characteristics.

Both evaluators ranked models similarly at the aggregate level. Evaluator 1 ranked fine-tuned ArabianGPT-08B-V2 highest (means ≈ 4.64–4.65) and fine-tuned AraGPT2-mega lowest (means ≈ 4.19–4.20), aligning with automatic evaluation results. Evaluator 2's rankings showed more nuanced distinctions: Llama-3-8B-Instruct highest (means ≈ 4.05–4.12),

followed closely by AceGPT-7B-chat (means ≈ 4.03–4.08) and ArabianGPT-08B-V2 (means ≈ 4.02–4.03), with AraGPT2-mega lowest (means ≈ 3.85–3.92). Despite these differences, the moderate to strong positive correlations indicate that both evaluators perceived similar quality patterns across models.

**Methodological considerations.** The evaluation protocol implemented in this study provides valuable lessons for future research in Arabic dialogue system assessment. While the evaluation demonstrated acceptable inter-rater reliability (fair to substantial agreement), several factors influenced the observed variation. The protocol lacked a formal calibration phase where evaluators could align their understanding of the rating scale through discussion of anchor examples. Evaluators worked independently without access to reference standards or guidelines for resolving ambiguous cases. The evaluation criteria (Fluency, Relevance, Diversity) may have been interpreted with different thresholds by each evaluator, particularly regarding distinctions between performance levels. The involvement of an author as an evaluator (Evaluator 1), while ensuring domain expertise, may have introduced different expectations compared to the external evaluator, as evidenced by the consistently higher ratings from Evaluator 1.

These observations inform recommendations for enhanced evaluation methodologies. The positive correlations between evaluators ($\rho = 0.424$–$0.759$) demonstrate that both perceived similar quality patterns, while the fair to substantial agreement levels ($\kappa = 0.229$–$0.739$) indicate acceptable but improvable consistency. The gradient in agreement levels, highest for the well-known baseline model (Llama-3) and lowest for the novel fine-tuned model (ArabianGPT), suggests that evaluator consensus may depend on model familiarity and linguistic characteristics. The near-perfect within-one agreement (0.999–1.000) indicates strong ordinal consistency, with evaluators rarely disagreeing on broad quality tiers despite differences in absolute ratings.

## Integrated assessment

The automatic metrics provide robust and consistent evidence supporting the effectiveness of fine-tuning pre-trained Arabic language models on synthetic data. Fine-tuned ArabianGPT-08B-V2's superior performance on the RAVEN metric (0.823) and strong within-model perplexity (9.4) demonstrate clear advantages in linguistic quality and response appropriateness. The ID vs. OOD analysis provides additional validation: the negligible performance degradation on held-out topic-country combinations (average <1% across all metrics) confirms that models learned generalizable conversational patterns rather than memorizing training examples. This robust cross-distribution consistency, validated through both automatic metrics and human evaluation, demonstrates the quality and diversity of the synthetic training data.

Human evaluation revealed convergent validity with automatic metrics across both evaluation paradigms. Both evaluators ranked ArabianGPT-08B-V2 favorably (Evaluator 1 ranked it highest; Evaluator 2 ranked it competitively), and critically, human evaluation scores showed the same pattern of OOD robustness as automatic metrics, with negligible degradation across distributions. The positive correlations between evaluators ($\rho = 0.424$–$0.759$) and fair to substantial agreement levels ($\kappa = 0.229$–$0.739$) indicate that, despite systematic rating differences, both evaluators perceived similar quality patterns across models and distributions. The convergence of automatic metrics (evaluated across thousands of conversations) and human evaluation (across multiple raters, criteria, and data distributions) provides comprehensive triangulated evidence for model quality and generalization capability.

## Architectural and training determinants of performance

The superior performance of fine-tuned ArabianGPT-08B-V2 (RAVEN: 0.823; Human: 4.34) despite its smaller parameter count (800M vs. 7–8B for baselines) can be attributed to three converging factors. First, its Arabic-exclusive pre-training on 14 + billion tokens ensures that 100% of model capacity is dedicated to Arabic linguistic patterns, avoiding the capacity dilution inherent in multilingual models like Llama-3-8B-Instruct, which distributes 15 trillion tokens across 30 + languages. Second, the AraNizer tokenizer (a purely Arabic BPE tokenizer with 64K vocabulary) efficiently encodes Arabic's complex morphology, whereas the baselines employ general-purpose tokenizers (tiktoken, LLaMA 2) optimized for English that may suboptimally

segment Arabic morphemes. Third, task-specific fine-tuning on the synthetic multi-turn dataset directly aligned model behavior with the evaluation criteria, compensating for architectural limitations such as the 1,024-token context window.

The convergence of RAVEN scores between fine-tuned AraGPT2-mega (0.780) and the instruction-tuned baselines (0.781–0.782) reveals an important interaction between base model characteristics and fine-tuning efficacy. Despite undergoing identical fine-tuning, AraGPT2-mega (released in 2020 using the GROVER architecture variant) did not achieve the same gains as ArabianGPT-08B-V2 (February 2024). This disparity likely reflects accumulated methodological advances in tokenization (AraNizer vs. GPT2TokenizerFast), training data curation, and architectural refinements that occurred between 2020 and 2024. Meanwhile, the competitive human evaluation scores of instruction-tuned baselines (4.18–4.22) without task-specific fine-tuning demonstrate that sophisticated alignment techniques (Reinforcement Learning from Human Feedback for Llama-3 and Reinforcement Learning from AI Feedback for AceGPT) produce broadly capable models, though their lower RAVEN scores indicate that general instruction-following capability does not fully substitute for task-specific adaptation in multi-turn Arabic dialogue.

These findings suggest that for specialized Arabic dialogue tasks, the alignment between training data and target task may outweigh raw model scale. The results indicate that smaller, Arabic-native models with task-specific fine-tuning can surpass larger multilingual alternatives, underscoring the importance of language-specific optimization, tokenization strategy, and task alignment as critical factors in Arabic LLM research.

## Conclusion

This study introduced a practical methodology for improving Arabic multi-turn conversational language models by leveraging synthetic data generated using an Arabic instruction-tuned LLM. The proposed approach involved prompt engineering, hyperparameter configuration, and large-scale dialogue generation covering a diverse set of topics and countries. The resulting dataset included 43,316 conversations and 608,052 utterances, offering a rich and diverse resource for fine-tuning pre-trained Arabic language models.

The evaluation of fine-tuned models demonstrated clear improvements on automatic metrics with robust generalization beyond the training distribution. Fine-tuned ArabianGPT-08B-V2 demonstrated strong within-model performance (Perplexity: 9.4) and achieved the highest cross-model RAVEN score (0.823), outperforming fine-tuned AraGPT2-mega and multilingual instruction-tuned baselines. Critically, comprehensive evaluation across in-distribution (5,970 conversations) and out-of-distribution (2,693 conversations from held-out topics and countries) test subsets revealed exceptional generalization, with negligible performance degradation (<1% average across RAVEN, perplexity, and human evaluation). This near-zero OOD degradation, validated through multiple evaluation methods, provides strong evidence that models learned generalizable conversational competence rather than dataset-specific patterns, confirming the quality and diversity of the synthetic training data.

Human evaluation demonstrated acceptable inter-rater reliability and yielded averaged scores that aligned with automatic metric rankings across both ID and OOD distributions. Inter-rater agreement ranged from fair to substantial (Cohen's $\kappa = 0.229$–$0.739$), with positive rank correlations (Spearman $\rho = 0.424$–$0.759$) and near-perfect ordinal agreement (within-one = 0.999–1.000). Exact agreement ranged from 0.375 to 0.834 across models and criteria. The systematic rating differences of 0.17–0.62 points between evaluators, while notable, remained within acceptable bounds for subjective assessment. Importantly, human evaluation scores demonstrated the same pattern of robust OOD generalization observed in automatic metrics, with negligible degradation on held-out distributions. The pattern of agreement levels, highest for the established baseline model (Llama-3: $\kappa = 0.635$–$0.739$) and lower for novel fine-tuned models (ArabianGPT: $\kappa = 0.229$–$0.247$), suggests that evaluator consensus depends partly on model familiarity. These findings highlight both the feasibility of human evaluation for Arabic dialogue systems and the need for continued methodological refinement.

This study makes several important contributions to Arabic NLP research. First, it demonstrates that synthetic data generation represents a viable and scalable approach for developing training resources for under-resourced languages, with

automatic metrics and OOD evaluation confirming both the quality improvements and robust generalization achievable through this methodology. Second, it introduces a novel evaluation benchmark specifically designed for Arabic multi-turn chatbots, combining automatic metrics with structured human evaluation and comprehensive OOD assessment using held-out topic-country combinations. The convergence of automatic and human metrics in demonstrating exceptional OOD generalization provides robust triangulated evidence for the effectiveness of this approach. Third, it provides transparent documentation of both successes and methodological considerations in evaluation, offering important lessons for the community regarding the complexities of assessing dialogue system quality and validating generalization capabilities.

The automatic metrics provide strong empirical support for fine-tuning pre-trained Arabic language models on synthetic conversational data, with OOD analysis confirming genuine generalization rather than memorization. Human evaluation showed convergent validity with these findings across both ID and OOD distributions. Future work should continue refining human evaluation methodologies by:

1. Recruiting a larger, more diverse pool of independent evaluators blind to model identity

2. Implementing structured calibration sessions with anchor examples spanning the full rating scale

3. Developing more detailed rubrics with concrete quality indicators for each criterion

4. Collecting multiple ratings per conversation to enable more robust statistical analysis

5. Incorporating qualitative feedback to understand the sources of rating differences

These improvements would further strengthen confidence in human evaluation results and provide clearer insight into the practical utility of fine-tuned Arabic dialogue models.

This methodology can be reproduced with emerging state-of-the-art Arabic LLMs, allowing for broader comparisons as more advanced models become available. Additionally, with access to greater computational resources, a wider range of pre-trained Arabic language models and multilingual language models could be fine-tuned and benchmarked, further strengthening the generalizability and impact of the approach. The demonstrated robustness of OOD generalization suggests that this synthetic data generation approach successfully captures fundamental aspects of Arabic multi-turn dialogue that transfer effectively to novel contexts. Future directions may also explore integrating Reinforcement Learning from Human Feedback (RLHF), expanding dialectal coverage, refining dialogue modeling techniques, and developing more reliable human evaluation frameworks that complement automatic metrics in assessing the nuanced qualities of conversational Arabic AI systems. By addressing both the technical challenges of model development and the methodological challenges of evaluation, including robust validation of generalization capabilities, the field can advance toward culturally and linguistically aligned dialogue systems that serve Arabic-speaking communities effectively across diverse topics and contexts.

## Supporting information

**S1 Text. English to Arabic translations.**
(DOCX)

**S1 Checklist. PLOSOne human subjects research checklist.**
(DOCX)

## Author contributions

**Conceptualization:** Ahmed Mahmoud Misbah.

**Data curation:** Ahmed Mahmoud Misbah.

**Formal analysis:** Ahmed Mahmoud Misbah.

**Investigation:** Ahmed Mahmoud Misbah.

**Methodology:** Ahmed Mahmoud Misbah.

**Resources:** Ahmed Mahmoud Misbah.

**Software:** Ahmed Mahmoud Misbah.

**Supervision:** Mohamed Farouk, Mustafa AbdulAzim.

**Validation:** Ahmed Mahmoud Misbah.

**Visualization:** Ahmed Mahmoud Misbah.

**Writing – original draft:** Ahmed Mahmoud Misbah.

**Writing – review & editing:** Mustafa AbdulAzim.

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
