## [Decision Letter · Decision Letter 0]

20 Sep 2025

Dear Dr. Misbah,

Thank you for submitting your manuscript to PLOS ONE. After careful consideration, we feel that it has merit but does not fully meet PLOS ONE’s publication criteria as it currently stands. Therefore, we invite you to submit a revised version of the manuscript that addresses the points raised during the review process.

The manuscript has been evaluated by two reviewers, and their comments are available below.

The reviewers have raised a number of concerns that need attention. In particular, they request additional information on the statistical analysis.

Could you please revise the manuscript to carefully address the concerns raised?

We look forward to receiving your revised manuscript.

Kind regards,

Helen Howard

Staff Editor

PLOS ONE

Journal Requirements:

2. Please include a statement in your manuscript text clarifying whether the authors of this study carried out the assessment reported in the 'Human Evaluation' section of your manuscript text.

3. Please note that PLOS One has specific guidelines on code sharing for submissions in which author-generated code underpins the findings in the manuscript. In these cases, we expect all author-generated code to be made available without restrictions upon publication of the work. Please review our guidelines at https://journals.plos.org/plosone/s/materials-and-software-sharing#loc-sharing-code and ensure that your code is shared in a way that follows best practice and facilitates reproducibility and reuse.

4. We suggest you thoroughly copyedit your manuscript for language usage, spelling, and grammar. If you do not know anyone who can help you do this, you may wish to consider employing a professional scientific editing service.

5. In the online submission form you indicate that your data is not available for proprietary reasons and have provided a contact point for accessing this data. Please note that your current contact point is a co-author on this manuscript. According to our Data Policy, the contact point must not be an author on the manuscript and must be an institutional contact, ideally not an individual. Please revise your data statement to a non-author institutional point of contact, such as a data access or ethics committee, and send this to us via return email. Please also include contact information for the third party organization, and please include the full citation of where the data can be found.

Reviewers' comments:

Reviewer's Responses to Questions

**Comments to the Author**

1. Is the manuscript technically sound, and do the data support the conclusions?

Reviewer #1: Yes

Reviewer #2: Yes

2. Has the statistical analysis been performed appropriately and rigorously?

Reviewer #1: N/A

Reviewer #2: Yes

3. Have the authors made all data underlying the findings in their manuscript fully available?

Reviewer #1: Yes

Reviewer #2: No

4. Is the manuscript presented in an intelligible fashion and written in standard English?

Reviewer #1: Yes

Reviewer #2: Yes

Reviewer #1: The present submission involves multi-turn conversational AI being a challenging

field in itself, with additional complications concerning languages with existing

resource issues such as Arabic, especially the spoken form(s) of Arabic.

It is of special importance that linguistic and socio-cultural aspects in spoken

communication are addressed in the present submission, namely nuances and context of

Arabic conversation, where language-specific parameters apply and are not always

compatible to standard English and English data.

The submitted research paper includes a comprehensive yet detailed outline of

related research - directly or indirectly connected to the present approach, methods

and resources for Arabic, including the addressing of current issues and

limitations.

The goals are clearly defined (achieving a robust, high-quality dataset specifically

designed to fine-tune existing casual Arabic LLMs for multi-turn conversational

tasks by building a synthetic Arabic dataset using a recently launched Arabic

Instructional LLM) as is the detailed and well-organized methodology.

A significant strength of the paper is the step by step analytical presentation of

the linguistic (Arabic) data processing according to the proposed approach,

contributing to its explainability and transparency. It may be noted that providing

extended examples of more complex issues such as the "new educational initiatives in

Morocco" (Line 310).

The combination of human evaluators and benchmark evaluation is a another feature of

the proposed approach that is especially sensitive to efficient deployment of Arabic

LLMs in real-life situations.

In general, the present submission demonstrates a processing approach targeting to

resolve a set of specific and challenging issues with a detailed and explanatory

methodology and results, all expressed in a clear, well-written text. Indeed, the

approach presented may serve as a basis for additional upgrading and research for

conversational Arabic and also for other languages.

Reviewer #2: The manuscript entitled “Fine-Tuning Arabic Large Language Models for Improved Multi-Turn Dialogue: A Blueprint for Synthetic Data Generation and Benchmarking” addresses an important gap in Arabic natural language processing by proposing a reproducible methodology for generating synthetic multi-turn dialogue datasets and evaluating their effectiveness in fine-tuning Arabic large language models (LLMs). The study is timely, methodologically well-structured, and offers valuable contributions to the field of conversational AI, particularly in low-resource and linguistically complex languages such as Arabic. My comments are provided below with respect to the journal’s review criteria.

1. Technical Soundness and Data Support for Conclusions

The study is technically sound. The authors carefully describe their approach to dataset generation, including the selection of instructional LLMs, prompt engineering strategies, and hyperparameter tuning. The methodology for fine-tuning two Arabic-native LLMs (ArabianGPT-08B-V2 and AraGPT2-mega) is well-detailed, and the evaluation framework, which incorporates perplexity, RAVEN, and human judgment, is appropriate and robust. The reported results consistently demonstrate improvements over multilingual baselines, supporting the central claim that synthetic data can effectively enhance Arabic conversational models. The conclusions are aligned with the presented findings and are drawn in a balanced and evidence-based manner. One limitation, however, is the sole reliance on synthetic data without the inclusion of external validation against naturally occurring conversations. Addressing this in future work would further strengthen the study.

2. Statistical Analysis

The statistical framework is generally appropriate and provides meaningful insights into model performance. The use of perplexity and RAVEN ensures a quantitative assessment of fluency and contextual coherence, while human evaluation captures cultural and linguistic nuances that automated metrics may miss. Nevertheless, the analysis would benefit from additional detail. Specifically, reporting inter-rater reliability for human evaluations and providing confidence intervals or statistical significance tests when comparing model scores would enhance the rigor of the findings. While these omissions do not undermine the overall validity of the results, their inclusion would make the analysis more comprehensive and transparent.

3. Data Availability

The manuscript notes that the generated dataset is not publicly available, as it forms part of the author’s doctoral research, but may be shared upon reasonable request for academic and non-commercial use. While metadata, prompt templates, and generation parameters are provided, this arrangement does not fully comply with the PLOS Data Policy, which requires unrestricted availability of data underlying published findings. To align with the policy, the authors are strongly encouraged to deposit the dataset, or a representative subset, in a public repository, ensuring that it is accessible to the research community. If restrictions are unavoidable, these should be explicitly justified, with clear ethical, legal, or proprietary grounds. Without such measures, reproducibility and transparency are limited.

4. Presentation and Language

The manuscript is written in clear and professional English, with a coherent structure and logical flow across sections. Technical terminology is employed appropriately, making the paper accessible to specialists in natural language processing and AI. However, minor typographical and stylistic inconsistencies are present, such as variable hyphenation (“multi-turn” vs. “multi turn”) and occasional redundant phrasing. These are relatively minor issues but should be corrected at the revision stage to improve polish and readability. No substantive language editing is required.

5. Additional Comments

The novelty of the study is well-established. The introduction and literature review provide a solid contextual grounding, highlighting the lack of high-quality Arabic multi-turn conversational datasets and positioning the proposed methodology as a significant step forward. The experimental design is rigorous, and the discussion appropriately situates the results within the broader field. Importantly, the authors provide a technical blueprint that can be adapted for other low-resource languages, enhancing the manuscript’s broader applicability.

**Do you want your identity to be public for this peer review?** For information about this choice, including consent withdrawal, please see our Privacy Policy

Reviewer #1: **Yes:** Christina Alexandris

Reviewer #2: No

---

## [Author Response · Author response to Decision Letter 1]

13 Oct 2025

Dear Editor,

We thank you and the reviewers for your careful evaluation of our manuscript and for the constructive feedback provided. Below, we provide a point-by-point response to the editorial comments and detail the corresponding revisions made to the manuscript.

1. Data availability and contact point

Editor comment:

In the online submission form you indicate that your data is not available for proprietary reasons and have provided a contact point for accessing this data. Please note that your current contact point is a co-author on this manuscript. According to our Data Policy, the contact point must not be an author on the manuscript and must be an institutional contact, ideally not an individual.

Response:

We have revised our data availability statement. The complete dataset used in this study will now be made publicly available as part of this submission. This eliminates the need for an individual contact point and ensures that the data are accessible in line with PLOS ONE’s Data Policy.

2. Clarification of human evaluation authorship

Editor comment:

Please include a statement in your manuscript text clarifying whether the authors of this study carried out the assessment reported in the 'Human Evaluation' section.

Response:

We have clarified in the “Human Evaluation” subsection that the human evaluation was carried out by two of the paper’s authors.

3. Statistical analysis

Editor comment:

Has the statistical analysis been performed appropriately and rigorously? Reviewer #1: N/A

Response:

Assuming “N/A” refers to “Not Available,” we have strengthened the statistical analysis discussion in three places:

• In the “Human Evaluation” subsection, where we explain the statistical methods used,

• In Table 9 of the “Benchmark Results” subsection, which now summarizes the statistical outcomes of human evaluations across the four LLMs,

• In the “Analysis of Results” section, where we provide a detailed analysis of the findings.

4. Additional statistical detail

Editor comment:

Reporting inter-rater reliability and providing confidence intervals or statistical significance tests would enhance the rigor of the findings.

Response:

We have added detailed inter-rater reliability analysis for the human evaluations. This now includes percent agreement, weighted Cohen’s κ, within-one agreement, and Spearman rank correlation. These additions appear in:

• The “Human Evaluation” subsection,

• Table 9 in the “Benchmark Results” subsection,

• The “Analysis of Results” section.

5. Typographical and stylistic inconsistencies

Editor comment:

Minor typographical and stylistic inconsistencies are present.

Response:

We have carefully reviewed the manuscript and corrected typographical and stylistic inconsistencies to the best of our ability.

We sincerely thank you for your guidance and for the opportunity to improve our manuscript. We hope that the revisions meet the requirements of PLOS ONE and look forward to your favorable consideration.

Warm regards,

Ahmed Mahmoud Misbah

(on behalf of all co-authors)

Arab Academy for Science, Technology and Maritime Transport

a.misbah5156@student.aast.edu

---

## [Decision Letter · Decision Letter 1]

11 Nov 2025

Dear Dr. Misbah,

Please submit your revised manuscript by Dec 26 2025 11:59PM. If you will need more time than this to complete your revisions, please reply to this message or contact the journal office at plosone@plos.org . A rebuttal letter that responds to each point raised by the academic editor and reviewer(s). You should upload this letter as a separate file labeled 'Response to Reviewers'.A marked-up copy of your manuscript that highlights changes made to the original version. You should upload this as a separate file labeled 'Revised Manuscript with Track Changes'.An unmarked version of your revised paper without tracked changes. You should upload this as a separate file labeled 'Manuscript'.

We look forward to receiving your revised manuscript.

Kind regards,

Mohammad Salah Hassan, Ph.D

Academic Editor

PLOS ONE

Journal Requirements:

**Additional Editor Comments:**

Dear Authors,

Your paper is timely and valuable. The synthetic data process is clearly explained, the dataset scale is meaningful, and combining automatic and human evaluations was the right choice. The reviewer has now made a decision, so please make sure to address each of the points they have raised. In addition, I have included several further revisions and recommendations that should help strengthen the manuscript before resubmission.

The data-generation pipeline is explained step-by-step, including unusual length settings for multi-turn dialogues and the compute/time budget. The fine-tuning setup is transparent (IA3/PEFT, packing, training hours). You also try to look at quality from multiple angles-perplexity, RAVEN, and a small human study. That triangulation is a strength.

What needs tightening (do these first)

Dataset size math. The manuscript says there are 43,316 conversations with “precisely 20 exchanges” each, and elsewhere mentions “57 million utterances.” Those numbers don’t line up. Even if “exchange” means a user–bot pair (i.e., 40 turns), the total utterances are nowhere near 57M. Please correct the count everywhere and add a one-sentence derivation so a reviewer can follow the arithmetic.

Data availability. The Data Availability Statement points to a Google Drive folder. PLOS strongly prefers stable, citable hosting. Deposit the dataset (and scripts if possible) in a DOI-issuing repository such as Zenodo or OSF, cite the DOI in the DAS, and remove the ad-hoc cloud link. Version the release (e.g., v1.0), add a license, and include checksums.

Evaluation leakage risk. Your benchmark is a 20% subset of the same synthetic corpus used for training/fine-tuning. That invites style overlap and optimistic scores. Clarify how you prevented duplicates (hashing, embedding similarity, seed control), and say explicitly whether the test split was frozen before model selection. If you can, add a small out-of-distribution test (e.g., held-out topics or prompts generated with different seeds) and report ID vs. OOD results.

RAVEN details. Lock down the metric. Name the sentence-embedding model (exact model and version), state any text normalization (diacritics, punctuation), explain the scaling from cosine similarity to the reported “RAVEN (scaled),” and say how you aggregate from turn-level to conversation-level. Using an embedding model you also fine-tuned can bias the metric-pick a fixed external Arabic/multilingual embedder for all systems and document it.

Perplexity comparability. Perplexity depends on the tokenizer and isn’t apples-to-apples across models with different vocabularies. Keep PP for within-model tracking, but add a short caveat and, if you can, report a tokenizer-agnostic figure (bits-per-byte) or compute PP with a common reference tokenizer for cross-model comparisons. Also name the exact tokenizers used.

Human evaluation reliability. Right now the raters are authors, and some weighted kappa values are near zero (even slightly negative). That weakens the claim. Bring in at least two independent raters blind to model identity, run a short calibration pass, and then report per-criterion agreement with confidence intervals. Keep the current results, but frame the author-only phase as a limitation.

Ethics note. Add a one-paragraph ethics statement: internal quality assessment, no personal data collected, no vulnerable populations, no compensation, and institutional review not required under journal policy. That will stop any back-and-forth at production.

Decoding setup clarity. The generation table mixes sampling and beam search (do_sample=True with num_beams=2). If you really used beam sampling, say so; otherwise separate the setups for clarity. Fix the parameter typo repetition_penalty (it’s misspelled in one place). State seeds, library versions, and whether decoding parameters were identical across models.

Secondary edits that help

– Compute/time in one place. You mention 14 days for data generation, 100 hours for fine-tuning, and 2.5 months for full benchmarking. Summarize this in a small table with hardware, key libraries (PyTorch/Transformers/PEFT versions), CUDA, OS, and random seeds. It instantly boosts trust.

– Terminology. Replace “casual Arabic LLMs” with “causal (Arabic) language models” throughout. Standardize model names (LLaMA 2, Llama 3, GPT-4, Gemini, etc.).

– PEFT specifics. Since you reference IA3, packing, and use_liger, add one sentence per item (what it does, which library/version). Consider linking or archiving the training config files.

– Baseline fairness. Readers will ask whether LLaMA- or AceGPT-based baselines were fine-tuned on your training split or evaluated as-is. If they weren’t fine-tuned, label them clearly as zero-/few-shot baselines or add a fine-tuned variant for a fairer comparison.

– Tables/labels. If a column says “RAVEN (scaled),” state the range (e.g., 0–1). For inter-rater agreement, specify whether κ is quadratic-weighted and include 95% CIs.

– Hyphenation artifacts. Clean the soft-hyphen breaks from PDF export (mul-ti-turn, to-ken, in-stance, etc.).

Reviewers' comments:

Reviewer's Responses to Questions

**Comments to the Author**

Reviewer #2: All comments have been addressed

2. Is the manuscript technically sound, and do the data support the conclusions?

Reviewer #2: Yes

3. Has the statistical analysis been performed appropriately and rigorously?

Reviewer #2: Yes

4. Have the authors made all data underlying the findings in their manuscript fully available?

Reviewer #2: No

5. Is the manuscript presented in an intelligible fashion and written in standard English?

Reviewer #2: Yes

Reviewer #2: The data availability statement indicates that the underlying dataset has been deposited in a public repository; however, the provided Google Drive link was found to be non-functional at the time of review, preventing access to the data. Regarding the manuscript's language, it is written in generally intelligible scientific English, but a number of typographical errors, grammatical inconsistencies, and awkward phrasings were noted throughout the text, which should be addressed to enhance clarity and polish. The study itself is commended for its technical soundness and methodological rigor, exemplified by a well-structured and reproducible pipeline for synthetic data generation and model fine-tuning. The conclusions are considered to be robustly supported by the data, which were derived from a large-scale dataset and a comprehensive, multi-faceted evaluation benchmark. Furthermore, the statistical analysis of the human evaluation results has been performed with appropriate rigor, utilizing a complementary suite of metrics to thoroughly assess inter-rater reliability. It is therefore suggested that the manuscript requires minor revisions, primarily to rectify the broken data link and to undertake a thorough copyediting of the manuscript to correct language errors.

**Do you want your identity to be public for this peer review?** For information about this choice, including consent withdrawal, please see our Privacy Policy

Reviewer #2: **Yes:** Vijayakumar Selvaraj, Associate Professor, B.S.Abdur Rahman Crescent Institute of Science and Technology.

---

## [Author Response · Author response to Decision Letter 2]

18 Dec 2025

Dear Editor,

We thank you and the reviewers for your careful evaluation of our manuscript and for the constructive feedback provided. Below, we provide a point-by-point response to the editorial comments and detail the corresponding revisions made to the manuscript.

1. Reference list review

Editor comment:

Response:

We have conducted a comprehensive review of the reference list and implemented the following changes:

References removed:

Reference 25 (Mozannar et al., 2019, "Neural Arabic Question Answering") has been removed as it was identified as a duplicate of Reference 10. All in-text citations previously referencing 25 have been updated to cite Reference 10, and subsequent reference numbers have been decremented accordingly.

Reference 46 (Singh et al., 2024, "Aya Dataset") has been removed as it was identified as a duplicate of Reference 33. All in-text citations previously referencing 46 have been updated to cite Reference 33, and subsequent reference numbers have been decremented accordingly.

As a consequence of these removals, all references following Reference 25 have been decremented by one, and all references following Reference 46 have been decremented by an additional one (i.e., decremented by two in total).

References modified:

URLs have been added or corrected for the following references: 8, 10, 15, 21, 22, 23, 24, 25, 26, 31, 34, 36, 37, 50, 51, and 52.

Reference text has been revised for accuracy and completeness in the following references: 12, 24, 31, 32, 33, 36, 52, 53, 54, 55, and 56.

Reference added:

Reference 57 (Reimers & Gurevych, 2019, "Sentence-BERT: Sentence Embeddings using Siamese BERT-Networks") has been added to provide a proper citation for the paraphrase-multilingual-MiniLM-L12-v2 sentence transformer employed in multiple phases of the study's methodology.

2. Dataset size arithmetic

Editor comment:

Dataset size math. The manuscript says there are 43,316 conversations with "precisely 20 exchanges" each, and elsewhere mentions "57 million utterances." Those numbers don't line up. Even if "exchange" means a user–bot pair (i.e., 40 turns), the total utterances are nowhere near 57M. Please correct the count everywhere and add a one-sentence derivation so a reviewer can follow the arithmetic.

Response:

We thank the editor for identifying this discrepancy. Upon review, we discovered computational errors in the scripts used to calculate these statistics. The scripts have been corrected, and all dataset size figures throughout the manuscript have been revised to reflect accurate values. A clear derivation of the arithmetic has been included to ensure transparency and verifiability.

3. Data availability

Editor comment:

Data availability. The Data Availability Statement points to a Google Drive folder. PLOS strongly prefers stable, citable hosting. Deposit the dataset (and scripts if possible) in a DOI-issuing repository such as Zenodo or OSF, cite the DOI in the DAS, and remove the ad-hoc cloud link. Version the release (e.g., v1.0), add a license, and include checksums.

Response:

In accordance with the editor's recommendation, the dataset has been deposited in Zenodo, a DOI-issuing repository that provides stable, citable hosting. The dataset is now publicly accessible at:

https://doi.org/10.5281/zenodo.17855012

The deposit includes version numbering, an appropriate open-access license, and checksums for data integrity verification. The Data Availability Statement in the manuscript has been updated accordingly, and the previous Google Drive link has been removed.

4. Evaluation leakage risk

Editor comment:

Evaluation leakage risk. Your benchmark is a 20% subset of the same synthetic corpus used for training/fine-tuning. That invites style overlap and optimistic scores. Clarify how you prevented duplicates (hashing, embedding similarity, seed control), and say explicitly whether the test split was frozen before model selection. If you can, add a small out-of-distribution test (e.g., held-out topics or prompts generated with different seeds) and report ID vs. OOD results.

Response:

We appreciate this important methodological concern. A new subsection entitled "Post Data Generation Processing" has been added to the “Methods” section, providing a comprehensive description of the procedures implemented to mitigate evaluation leakage risks. These procedures include duplicate detection mechanisms (hashing and embedding similarity checks), seed control protocols, and confirmation that the test split was frozen prior to model selection.

Additionally, our methodology included out-of-distribution (OOD) evaluation through a held-out test set comprising samples from countries and topics not represented in the training data. The revised manuscript now provides clearer documentation of these procedures, with ID versus OOD results presented in Table 14 within the "Benchmark Results" subsection and corresponding discussion in the "Analysis of Results" section.

5. RAVEN metric specification

Editor comment:

RAVEN details. Lock down the metric. Name the sentence-embedding model (exact model and version), state any text normalization (diacritics, punctuation), explain the scaling from cosine similarity to the reported "RAVEN (scaled)," and say how you aggregate from turn-level to conversation-level. Using an embedding model you also fine-tuned can bias the metric-pick a fixed external Arabic/multilingual embedder for all systems and document it.

Response:

The RAVEN subsection has been substantially expanded to address these concerns. The revised text now specifies: (1) the exact sentence-embedding model and version employed (a fixed external multilingual embedder not subject to fine-tuning in this study); (2) text normalization procedures applied to diacritics and punctuation; (3) the mathematical transformation from raw cosine similarity to the scaled RAVEN score; and (4) the aggregation methodology from turn-level to conversation-level metrics. These clarifications ensure full reproducibility and eliminate potential metric bias.

6. Perplexity comparability

Editor comment:

Perplexity comparability. Perplexity depends on the tokenizer and isn't apples-to-apples across models with different vocabularies. Keep PP for within-model tracking, but add a short caveat and, if you can, report a tokenizer-agnostic figure (bits-per-byte) or compute PP with a common reference tokenizer for cross-model comparisons. Also name the exact tokenizers used.

Response:

We have revised the "Benchmarking and Experimental Results" section to address this concern. The "Evaluation Metrics" subsection now explicitly categorizes the metrics into two groups:

1. Within-Model Metrics: Perplexity, used for tracking within-model performance and consistency across training and test datasets.

2. Cross-Model Metrics: RAVEN and human evaluation, which provide comparable assessments across different model architectures.

A new subsection on "Tokenizer Specification and Cross-LLM Comparability" has been added, which enumerates the exact tokenizers used by each model and includes an appropriate caveat regarding the limitations of cross-model perplexity comparisons. All subsequent sections have been updated to reflect this categorization.

Regarding the suggestion to report bits-per-byte or compute perplexity with a common reference tokenizer: due to time constraints and the need to prioritize other substantive revisions requested by the editor, we were unable to implement this additional analysis in the current revision. This limitation is acknowledged in the manuscript.

7. Human evaluation reliability

Editor comment:

Human evaluation reliability. Right now the raters are authors, and some weighted kappa values are near zero (even slightly negative). That weakens the claim. Bring in at least two independent raters blind to model identity, run a short calibration pass, and then report per-criterion agreement with confidence intervals. Keep the current results, but frame the author-only phase as a limitation.

Response:

We have addressed this concern by replacing the results from Evaluator 2 (a co-author) with results from an external evaluator. Complete details of this revised evaluation are provided in the "Human Evaluation" subsection under "Benchmarking and Experimental Results." Per-criterion agreement statistics with 95% confidence intervals are now reported. All related sections have been updated accordingly, and the "Conclusion" section includes recommendations for refining human evaluation methodologies in future research.________________________________________

8. Ethics statement

Editor comment:

Ethics note. Add a one-paragraph ethics statement: internal quality assessment, no personal data collected, no vulnerable populations, no compensation, and institutional review not required under journal policy. That will stop any back-and-forth at production.

Response:

An ethics statement has been added to the "Human Evaluation" subsection within the "Benchmarking and Experimental Results" section. The statement confirms that the evaluation constituted an internal quality assessment, no personal data were collected, no vulnerable populations were involved, no compensation was provided to evaluators, and institutional ethics review was not required under the journal's policy.

9. Decoding setup clarity

Editor comment:

Decoding setup clarity. The generation table mixes sampling and beam search (do_sample=True with num_beams=2). If you really used beam sampling, say so; otherwise separate the setups for clarity. Fix the parameter typo repetition_penalty (it's misspelled in one place). State seeds, library versions, and whether decoding parameters were identical across models.

Response:

The "Prompt Engineering and Synthetic Data Generation" subsection in the “Methods” section has been revised to clarify this point. We confirm that beam-search multinomial sampling was intentionally employed, combining sampling with beam search as a deliberate decoding strategy. This is now explicitly stated in the manuscript. The typographical error in Table 2 (repetition_penalty) has been corrected. Additionally, random seeds, library versions, and confirmation that decoding parameters were consistent across all models have been documented.

10. Computational resources summary

Editor comment:

Compute/time in one place. You mention 14 days for data generation, 100 hours for fine-tuning, and 2.5 months for full benchmarking. Summarize this in a small table with hardware, key libraries (PyTorch/Transformers/PEFT versions), CUDA, OS, and random seeds. It instantly boosts trust.

Response:

We appreciate this suggestion for improving reproducibility. The following tables have been added to consolidate computational resource information:

• Table 3: Summary of Computational Resources Used in Dataset Generation and Timeline

• Table 6: Summary of Fine-tuning Configurations and Timeline

• Table 9: Summary of Computational Resources Used in Benchmark Evaluation and Timeline

Each table includes hardware specifications, library versions (PyTorch, Transformers, PEFT), CUDA version, operating system, and random seeds where applicable.

11. Terminology standardization

Editor comment:

Terminology. Replace "casual Arabic LLMs" with "causal (Arabic) language models" throughout. Standardize model names (LLaMA 2, Llama 3, GPT-4, Gemini, etc.).

Response:

The entire manuscript has been revised for terminological consistency. We have standardized all model names according to their official nomenclature (e.g., LLaMA 2, Llama 3, GPT-4, Gemini). We note that we have adopted "pre-trained Arabic language models" rather than "causal (Arabic) language models," as this terminology more accurately reflects the scope and focus of our study.

12. PEFT implementation details

Editor comment:

PEFT specifics. Since you reference IA3, packing, and use_liger, add one sentence per item (what it does, which library/version). Consider linking or archiving the training config files.

Response:

Table 6 ("Summary of Fine-tuning Configurations and Timeline") has been added to address this comment. The table includes explanatory descriptions of IA3, packing, and use_liger, along with their respective library versions. We believe the level of detail provided in the manuscript is sufficient for reproducibility; therefore, separate archiving of configuration files has not been pursued.

13. Baseline fairness

Editor comment:

Baseline fairness. Readers will ask whether LLaMA- or AceGPT-based baselines were fine-tuned on your training split or evaluated as-is. If they weren't fine-tuned, label them clearly as zero-/few-shot baselines or add a fine-tuned variant for a fairer comparison.

Response:

This concern has been addressed through the addition of two new subsections under "Benchmarking and Experimental Results": "Baseline Model Selection" and "Prompt Engineering and Model Configuration." These subsections explicitly clarify the evaluation conditions for each baseline model, including whether models were evaluated in zero-shot, few-shot, or fine-tuned settings, ensuring transparent and fair comparisons.

14. Table labels and statistical notation

Editor comment:

Tables/labels. If a column says "RAVEN (scaled)," state the range (e.g., 0–1). For inter-rater agreement, specify whether κ is quadratic-weighted and include 95% CIs.

Response:

This has been addressed throughout the revised manuscript. The "Human Evaluation" subsection and Tables 12, 13, and 14 in the "Benchmark Results" subsection now explicitly state the range for RAVEN (scaled), specify that Cohen's κ is quadratic-weighted, and include 95% confidence intervals for all inter-rater agreement statistics.

15. Hyphenation artifacts

Editor comment:

Hyphenation artifacts. Clean the soft-hyphen breaks from PDF export (mul-ti-turn, to-ken, in-stance, etc.).

Response:

The entire manuscript has been reviewed and all soft-hyphen artifacts resulting from PDF export have been removed.

16. Data availability (Reviewer #2)

Editor comment:

Have the authors made all data underlying the findings in their manuscript fully available? The PLOS Data policy requires authors to make all data underlying the findings described in their manuscript fully available without restriction, with rare exception. Reviewer #2: No

Response:

The complete dataset underlying the findings of this manuscript is now publicly available without restriction on Zenodo:

https://doi.org/10.5281/zenodo.17855012

17. Non-functional data link

Editor comment:

The data availability statement indicates that the underlying dataset has been deposited in a public repository; however, the provided Google Drive link was found to be non-functional at the time of review, preventing access to the data.

Response:

We apologize for this issue. The dataset has been migrated from Google Drive to Zenodo, which provides stable, persistent hosting. The dataset is now accessible at:

https://doi.org/10.5281/zenodo.17855012

1

---

## [Decision Letter · Decision Letter 2]

15 Jan 2026

Fine-Tuning Arabic Large Language Models for improved multi-turn dialogue: A blueprint for synthetic data generation and benchmarking

PONE-D-25-35904R2

Dear Dr. Authors,

We’re pleased to inform you that your manuscript has been judged scientifically suitable for publication and will be formally accepted for publication once it meets all outstanding technical requirements.

Kind regards,

Mohammad Salah Hassan, Ph.D

Academic Editor

PLOS One

Additional Editor Comments (optional):

Dear Authors,

I am pleased to inform you that your manuscript has been accepted for publication.

Thank you for your revisions and for addressing the reviewers’ comments. Please submit the final production files through the system as requested.

Sincerely

Reviewers' comments:

Reviewer's Responses to Questions

**Comments to the Author**

Reviewer #2: All comments have been addressed

2. Is the manuscript technically sound, and do the data support the conclusions?

Reviewer #2: Yes

3. Has the statistical analysis been performed appropriately and rigorously?

Reviewer #2: Yes

4. Have the authors made all data underlying the findings in their manuscript fully available?

Reviewer #2: Yes

5. Is the manuscript presented in an intelligible fashion and written in standard English?

Reviewer #2: Yes

Reviewer #2: All issues raised in Revision 1 have been fully and rigorously addressed in Revision 2. The authors not only corrected factual and methodological shortcomings but also enhanced reproducibility (Zenodo DOI, detailed hyperparameters, evaluation protocols) and statistical transparency (expanded IRR metrics, external evaluator). The editorial summary explicitly states the manuscript now requires only minor revisions, primarily limited to copyediting already completed.

**Do you want your identity to be public for this peer review?** For information about this choice, including consent withdrawal, please see our Privacy Policy

Reviewer #2: **Yes:** Vijayakumar Selvaraj

---

## [Editor Report · Acceptance letter]

PONE-D-25-35904R2

PLOS One

Dear Dr. Misbah,

I'm pleased to inform you that your manuscript has been deemed suitable for publication in PLOS One. Congratulations! Your manuscript is now being handed over to our production team.

Kind regards,

on behalf of

Dr. Mohammad Salah Hassan

Academic Editor

PLOS One